# Nonlinear optical vector processing using linear silicon photonic circuits for 50 Gb/s memory and string similarity functions

T. Moschos [1,2] ✉, C. Pappas [1,2], S. Kovaios [1,2], I. Roumpos[1,2], A. Prapas [1,2], A. Tsakyridis [1,2], M. Moralis-Pegios [1,2], C. Vagionas [1,2], Y. London [3], B. Tossoun [4], T. Van Vaerenbergh [3] & N. Pleros[1,2]

The need for high-speed, energy-efficient computing in machine learning and communications necessitates innovations beyond conventional electronics to sustain computational power advances without requiring prohibitive energy amounts. Photonics have emerged in various applications, demonstrating significant highlights in optical linear transformations, while if successfully employed can also be used in nonlinear processes and matching functionalities. Towards this we demonstrate nonlinear optical vector processing in the form of hamming distance calculations and content addressable memory banks, using linear silicon photonic circuits at the high-speed of 50 Gb/s, enabling pattern matching and look-up operations. The processor employs a 4×4 crossbar architecture utilizing silicon germanium modulators computing hamming distance between 2-bit optical vectors. It achieves error-rates of ~$10^{-3}$ in binary/ternary content matching, improving state-of-the-art demonstration speeds by >2.5×. Scalability is enhanced through space-wavelength multiplexing via a wavelength-division multiplexing cell, experimentally demonstrated at 50 Gb/s, offering increased computational capacity with reduced insertion losses and power consumption.

The demand for high-speed and energy-efficient computing continues to rise with the proliferation of data-intensive applications, such as machine learning, genomic analysis, and real-time communication[1]. In this race, digital electronic systems comprise still the steam engine in today's computational landscape, though analog electronics are also gaining momentum as a viable alternative for energy efficient computations directly in the analog domain[2]. However, as data rates and dataset sizes continue to grow, electronic computing in both its digital and analog versions is facing diminishing returns due to the native physical constraints of electronic technology: the size and energy advantages of electronic circuitry are naturally counteracted by the speed and power limits of the electronic interconnects inside the circuits due to RC parasitic effects[3], with current electronic processors hardly exceeding GHz clock frequencies. These limitations indicate

that a radical shift from conventional electronic computing architectures towards novel hardware computing paradigms need to be realized.

In this direction, silicon photonics (SiPho) emerge as a promising candidate for penetrating the processing and compute domains to turn their well-known dominance in the interconnect sector into a profound advantage also in the computational segment. Migrating, however, into a light-enabled processor technology paradigms has to ensure the successful deployment of fundamental computing operations at both symbol- and vector-/string-level in the optical domain. In this realm, photonics have already shown an impressive potential in implementing universal linear transformations over analog optical vectors[4–8] facilitating critical matrix and tensor multiplier building blocks in application fields like quantum[9,10], neuromorphic[11–16] and

[1]Department of Informatics, Aristotle University of Thessaloniki, Thessaloniki, Greece. [2]Center for Interdisciplinary Research and Innovation, Thessaloniki, Greece. [3]Hewlett Packard Labs, Milpitas, CA, USA. [4]Hewlett Packard Labs, Santa Barbara, CA, USA. ✉e-mail: moschost@csd.auth.gr

microwave photonics[17,18]. However, a broad range of applications like pattern matching, error correction and similarity searches require nonlinear processing functions like comparison and distance calculation processes between strings. These necessitate typically the use of memory and register blocks that are certainly not yet among the strengths of the photonic circuitry[19,20]. The calculation of string metrics, like the well-known Hamming Distance (HD), provides critical quantitative information about string and vector similarity[21], with certain distance values leading often to completely new functional blocks. Content Addressable Memories (CAMs), for example, form a specialized yet highly important pattern matching application[22] that correspond to a zero HD and constitute a critical element in high-speed routers, address and database lookups, and associative search functions[23]. Although optical CAMs experienced a significant progress during the last decade[24–26] and raised expectations even for complete high-speed look-up operations via silicon photonic Microring Resonator- (MRRs)[27] or electro-absorption modulator-based (EAM) setups[28], the transition from the specific vector matching to the more generic nonlinear processes like vector distance calculation has still not identified a viable photonic implementation route that would broaden the computational portfolio of integrated optics.

In this paper we introduce, to the best of our knowledge, the nonlinear transformation of optical vectors by using a linear SiPho circuit and demonstrate HD calculation and CAM operation between binary optical strings at record-high data rates of 50 Gb/s. The processor relies on the recently introduced Crossbar (Xbar) architecture[5], configured as a 4 × 4 matrix arrangement and employing 56 GHz bandwidth silicon germanium (SiGe) EAMs as its core computational cells. Its performance was experimentally evaluated in calculating the HD between 2-bit optical vectors, presenting also successful operation as optical CAM and Ternary CAM (TCAM) memory bank for 2-bit

optical words with matching error-rates of ~$10^{-3}$ at operational speeds 2.5× higher than current state-of-the-art CAMs clock speed. Moreover, we introduce a viable roadmap for increasing computational capacity by using space-wavelength multiplexed memory cells, demonstrating experimentally a wavelength division multiplexing (WDM)-enabled architecture that encodes optical vectors across a 2D space-wavelength dimension and performs vector processing at 50 Gb/s.

## Results

### Hamming distance calculation using a silicon photonic crossbar integrated circuit

HD is a metric widely employed in coding theory and error correction codes[29], in information theory[30], in Hamming clustering methods for data clustering[31], as well as in bioinformatics for DNA sequencing[32], underscoring its wide-reaching impact across computational and scientific disciplines. It is a nonlinear operation defined as the number of positions at which the corresponding symbols (characters or bits) differ. An illustrative example of its application is depicted in Fig. 1a, where the HD metric is applied to two N-length binary vectors X and Y. Mathematically the HD operation is defined by the following equation:

$$HD(X,Y) = \sum_{i=1}^{N} L(X(i) \neq Y(i)) \tag{1}$$

with the function $L(X(i) \neq Y(i))$ indicating all the positions in the N-length vectors X and Y where their corresponding values differ. The most prominent implementation of the HD operator in digital electronics comprises a XOR-gate array, followed by a Full-Adder Tree, as schematically illustrated in Fig. 1b[33]. Implementing this nonlinear HD calculation process via linear transformations can be realized by

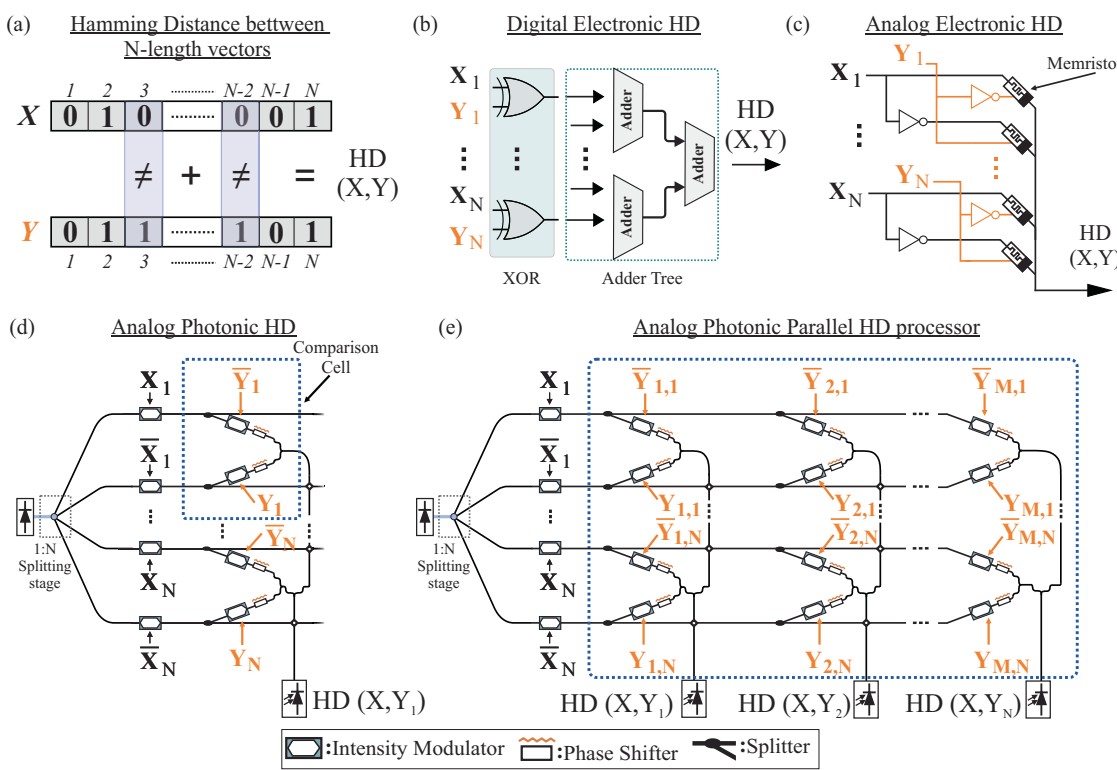

**Fig. 1 | Electronic and photonic implementations of N-length hamming distance processor architectures. a** Illustrative example of the HD metric, when applied to two different N-length binary vectors X and Y. **b** Digital electronic HD circuit. The presented implementation utilizes XOR comparison gates and an adder tree for N-symbol length comparison. **c** Analog electronic implementation of the HD operation using memristive elements. **d** Analog photonic HD circuit, based on amplitude modulators and coherent light summation, with the use of thermo-optic phase shifters. **e** Parallel analog photonic HD processor based on a coherent crossbar array layout, for N-length optical vector comparison operations. HD hamming distance, XOR Exclusive OR.

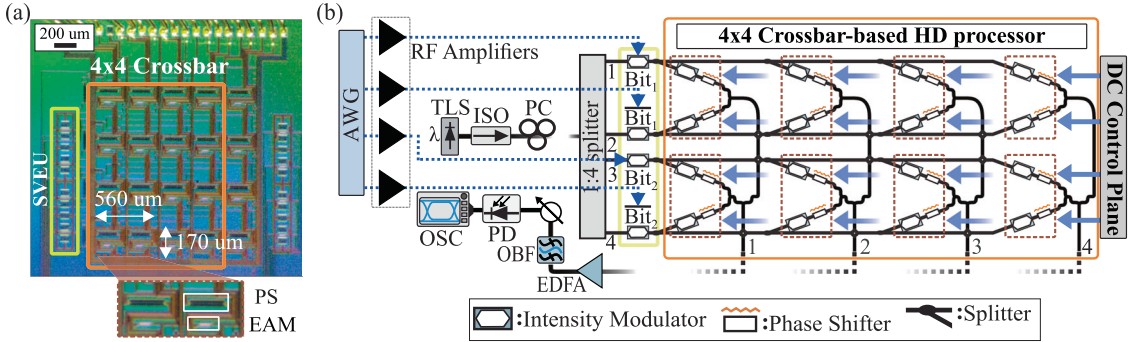

**Fig. 2 | Proposed integrated silicon photonic crossbar-based hamming distance processing unit. a** Microscope photo of the integrated 4 × 4 crossbar prototype comparison unit. Inset: Single crossbar comparison cell, consisting of two EAMs and two TO PSs. The search vector encoding unit and the 4 × 4 crossbar, are marked with yellow and orange rectangles, respectively. **b** Experimental testbed for the HD comparison operation at 20 and 50 Gb/s, highlighting the search vector encoding unit (yellow) and the 4 × 4 crossbar-based HD processor (orange), along with the respective comparison cells (brown). A DC control plane is employed for biasing the static EAMs and PSs located in the crossbar layout. EAM electro-absorption modulator, TO thermo-optic, PS phase-shifter, HD hamming distance.

doubling the vector dimensions to incorporate also the inverted vector values and applying a dot product operation, as has been initially demonstrated via a rather low-speed analog electronic HD circuitry, depicted in Fig. 1c[21]. In this layout, the actual and complementary values of array Y are programmed in a memristor array, while the X actual and complementary values are injected as voltage signals across the memristor array. The resulting output current is, due to Kirchhoff's law, directly correlated to the HD of the input X and Y vectors.

The employment of this mathematical concept and the transfer of the memristor-based architectural approach into the photonic domain can be realized by adopting an equivalent analog optical circuit, as depicted in Fig. 1d, e, by: (i) encoding the X actual and complementary vector values into a modulated optical light carrier (ii) encoding the Y actual and complementary values in the transmission values of optical amplitude modulators, which can be assumed to act as optical memory cells that store the corresponding transmission value, (iii) injecting the encoded X modulated light beams to the Y amplitude modulating elements and adding coherently the resulting optical fields, by utilizing thermo-optic phase shifters that ensure proper phase matching between the two branches. The comparison cell highlighted in Fig. 1d, comprises the basic building block of this comparison operation (details for its functionality are also provided in Supplementary Section 1). The basic analog photonic HD circuitry that performs a dot product operation between two optical vectors has been initially presented in ref. 34 and is shown in Fig. 1d. This can be expanded to a parallel HD processor by employing the coherent Xbar architecture, originally proposed in ref. 5 and depicted in Fig. 1e. In this layout the X actual and complementary values are equally broadcasted to the corresponding M Xbar columns using directional couplers, with properly selected splitting ratios. Each column of the Xbar is configured to hold different predefined/target vectors, with each comparison cell assigned to a distinct symbol value. Hence, the HD operation between the incoming X vector and the respective target #M Y Vectors of the Xbar array, is executed in parallel across all Xbar columns allowing up to M parallel HD calculations.

The functionality of the proposed analog photonic HD processor architecture was assessed via a 4 × 4 SiPho integrated Xbar layout that is capable of 2-bit vector distance processing, surpassing prior optical CAM implementations[27,28], which have demonstrated only 1-bit comparison operations. The photonic chip was fabricated in imec's SiPho platform, using PDK-ready components. Particularly, every computing cell of the Xbar prototype incorporates 50 µm long, 56 GHz Franz–Keldysh (FK) SiGe EAMs and 170 µm long thermo-optic phase shifters (TO-PSs). The deployed photonic integrated circuit (PIC) supports 2-bit search vectors, encoded through the search vector

encoding unit (SVEU), highlighted with the yellow rectangle in Fig. 2a, that utilizes four EAMs to generate two pairs of complementary search values. The encoded search information is then forwarded into the Xbar-based HD processor, with the corresponding chip area marked with an orange rectangle. The optical signals are distributed across all Xbar columns, which statically encode all the target vectors at every EAM column. The HD result ("match"/degree of "mismatch") between the search and target vectors is generated at each Xbar output column. The experimental testbed, established for the evaluation of the HD operations, is depicted in Fig. 2b. A tunable laser source (TLS) is employed to generate a continuous wave (CW) optical signal at 1563 nm which is injected into the 4 × 4 Xbar chip. A 4-channel arbitrary waveform generator (AWG) is used to generate the 2-bit electrical search vector. The first and third channel produces the $Bit_1$ and $Bit_2$, respectively, while the second and fourth channels generates their complementary values, i.e. $\bar{Bit}_1$ and $\bar{Bit}_2$. After electrical amplification, the electrical signals are fed into the input EAMs (yellow rectangle) to encode the optical SVEU. The target 2-bit vectors (i.e., "00", "01", "10" and "11") are implemented by setting the optical attenuation of EAMs located at each Xbar row. Thermo-optic phase shifters are deployed to ensure constructive interference at the output columns of the Xbar. Both EAMs and TO PSs are controlled by a multi-channel DC control plane, that consists of 16 programmable 2-channel DC power supplies. The output of each column is captured by an oscilloscope, after being amplified and detected by a photodiode. More details of the experimental setup can be found in methods section.

## Experimental validation of hamming distance calculation and ternary CAM memory bank at 50 Gb/s

The performance of the proposed parallel HD processor was experimentally assessed with 2-bit optical vectors encoded onto the SVEU and compared with respective 2-bit optical vectors stored onto the four Xbar columns. The case of HD = 0 designates an exact vector matching and corresponds to the match-line (ML) operation of a complete CAM memory bank, with the incoming optical vector representing the search word and the stored optical vectors the respective address words. The photonic comparison cells are used for representing all the memory states of a Binary CAM cell, i.e. "0", "1", but can also support the wildcard/ternary value "X" by storing a zero-transmission value to both EAMs within the comparison cell, allowing in this way for an extension to ternary CAM (TCAM) applications.

The HD and ML operations performed through the Xbar processor have been experimentally validated for search vector streams at 20 Gb/s and 50 Gb/s. Figure 3a, c illustrates the time sequences of the 2-bit optical vector within the first two rows of the figure for a 10-bit

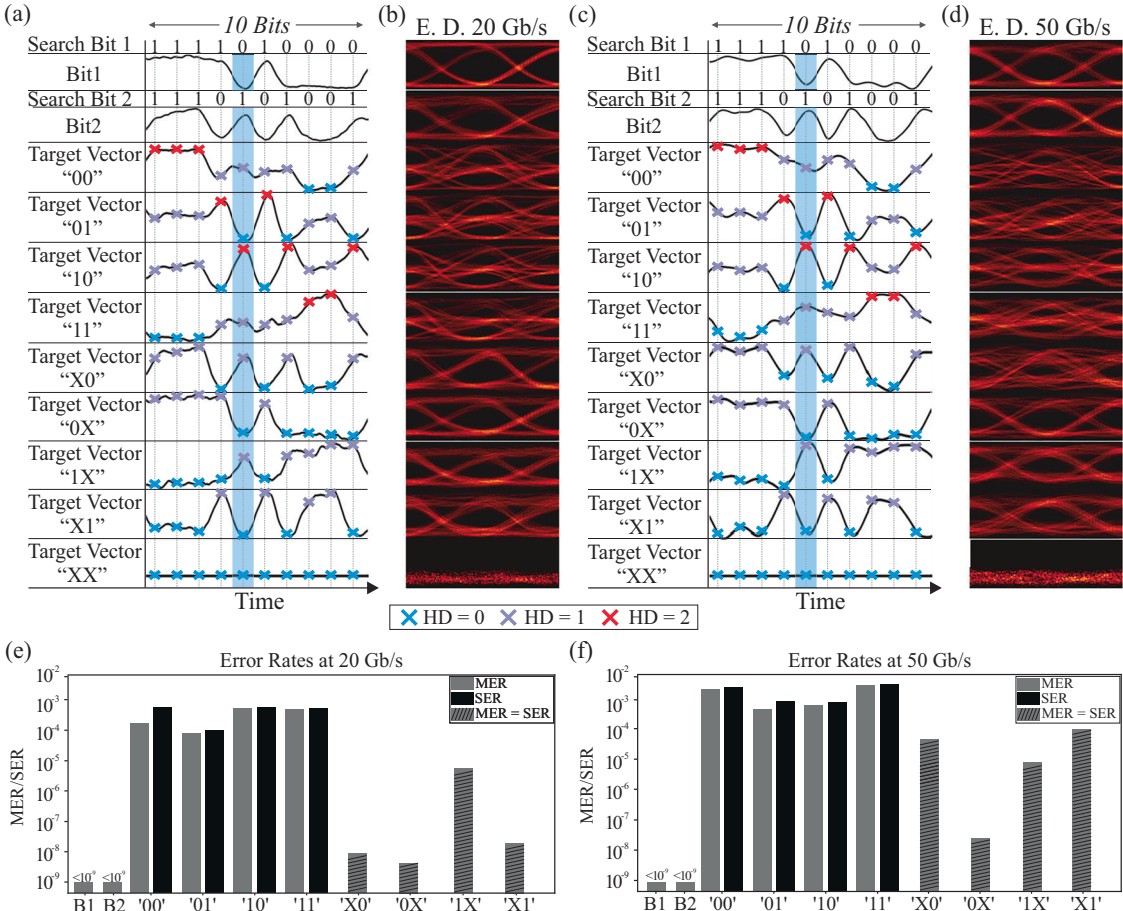

**Fig. 3 | Experimental results and error rate values for the validation of the 2-symbol vector comparison, at 20 and 50 Gb/s operational data rates.**
**a** Synchronized time traces (50 ps/div) showing 10 encoded search bits (Bit₁-Bit₂) along with the respective comparison outcomes with the target vector cases of "00"-"XX". **b** Corresponding eye diagrams at 20 Gb/s operation. **c** Synchronized time traces (20 ps/div) of 10 optically encoded search bits (Bit₁-Bit₂) and comparison contents. **d** Eye diagrams at 50 Gb/s operation. **e** Error-rates of all cases at 20 Gb/s. **f** Error rates when the processor operates at 50 Gb/s. MER match error rate, SER symbol error rate, ED eye diagrams.

time window at 20 Gb/s and 50 Gb/s operation, respectively, with the additional rows depicting the respective time sequences obtained at the Xbar column output port when the column stored content matches the designated target vector. The corresponding eye diagrams for every time sequence are shown in Fig. 3b and d, respectively. For all cases, the cross-shaped scatter points indicate the optimum sampling points, corresponding to one of the three possible output states of the HD operation i.e. HD = 0 (ML operation), HD = 1, and HD = 2. Particularly, the green highlighted area showcases the principle of operation for a search vector value of '01', that corresponds to Bit₁ and Bit₂ values of '0' and '1', respectively. A logical "match" state occurs, only when compared with the target vectors of "01", "0X", "X1" and "XX". As expected, this results in a "zero" output power (HD = 0), since the wildcard value "X" represents a "don't care state" leading to a logical "match" regardless of the corresponding search value. On the other hand, if the compared vectors have different values, the comparison operation indicates a logical "mismatch", resulting in an HD > 0. Specifically, if only one of the two search bits matches the content of the Xbar array, the HD is "1" (HD = 1). Conversely, if none of the search bits match the array values, the HD is "2" (HD = 2), as illustrated in the case of the "10" case time trace.

The performance of the 4×4 vector processor was assessed by calculating the projected symbol error rate (SER), when HD calculation is targeted, and the match error rate (MER) when optical CAM operation is intended, for each different target vector case. The SER

quantifies the HD performance, while the MER specifically refers to the error rate of the '0' level that corresponds solely to the matching functionality without accounting for the degree of mismatch (for more details see Supplementary Material Section 3). Figure 3e, f illustrates the calculated MER and SER when the Xbar HD processor operates at 20 Gb/s and 50 Gb/s, respectively. As can be observed, all error-rates are in the order of 10⁻³, comparable with the performance achieved by multi-symbol electronic HD and CAM computing cells[35–37]. The results indicate a degradation in both SER/MER measurements as the data rate increases, primarily due to noise arising from the limited frequency response of the deployed electro-optic components. Finally, it is worth mentioning that when the wildcard 'X' is included, the output signals are restricted to two levels and MER becomes equivalent to SER, resulting at the same time in improved performance compared to 3-level signals.

## WDM-enabled comparison cell architecture for scalable hamming distance processor layouts
In the previous section, we validated the capabilities of the coherent Xbar architecture in performing HD operations between vectors by leveraging spatially distributed comparison cells and the interference properties of a single coherent light beam. Synergizing the WDM capabilities of the Xbar architecture[38,39], with the HD architectural approach, can offer an additional axis of computational capacity at a reduced insertion loss budget (for more information see Supplementary Material

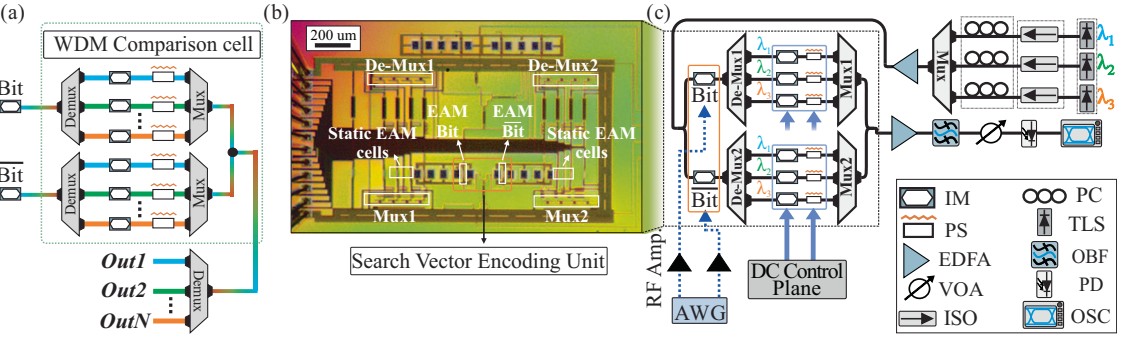

**Fig. 4 | Proposed integrated silicon photonic WDM crossbar-based comparison unit. a** Schematic diagram of the proposed WDM comparison cell, operating with N number of wavelength values. **b** Microscope image of the WDM-enabled silicon photonic HD processor. Highlighted: Search Bit and $\overline{Bit}$ EAMs (Search Vector Encoding Unit), comparison cell EAMs, Multiplexer and De-multiplexer modules. **c** Circuit schematic and experimental setup of the 3 wavelength ($\lambda_1$, $\lambda_2$ and $\lambda_3$) 2 × 1 HD processor. The orange highlighted area depicts the optical Search Vector Encoding Unit. WDM wavelength division multiplexing, HD hamming distance.

Section 4), thereby enhanced energy efficiency. Figure 4a illustrates the proposed WDM-enabled comparison cell. The actual and complementary search vector symbol (*Bit* and $\overline{Bit}$) are used to modulate two multi-wavelength light beams. The resulting modulated streams are demultiplexed to their wavelength constituents via a demultiplexer[40], with each wavelength propagating through an intensity modulator (IM) that encodes a value from the target vector. The resulting signals are subsequently multiplexed and coherently recombined on a wavelength basis via constructive interference, achieved using TO PSs. Finally, the output is demultiplexed, with each wavelength carrying the dot-product between the assigned symbol and the search bit value. Further information about the functionality of the WDM-based cell is presented in Supplementary Material Section 1. This WDM-enabled comparison cell facilitates the parallel execution of HD operations within a single output, where the number of wavelengths represents the level of parallelism. In order to validate the advantages of the WDM-enabled approach a $3\lambda - 2 \times 1$ comparison cell was fabricated using the same technological building blocks (i.e. EAM and TO-PS) as in the $4 \times 4$ Xbar. A microscope photo of the fabricated chip is depicted in Fig. 4b, while the circuit schematic of the WDM-enabled chip along with the experimental setup used to assess its performance in executing HD operations are presented in Fig. 4c. Additional information about the experimental testbed of the WDM-enabled HD chip can be found in the methods section. It is worth noting that our photonic integrated chip incorporates (de)multiplexers originally designed to support 4 different wavelengths. However, due to fabrication issues, only 3 wavelength channels were fully functional, while the fourth exhibited significant insertion losses. Therefore, we successfully demonstrated parallel operation of 3 wavelengths at up to 50 Gbps.

The experimental validation of the proposed WDM-enabled HD processor was conducted at data rates of 20 and 50 Gb/s, employing 3 different wavelengths. Figure 5a depicts indicative 10-bit long time traces at 20 Gb/s operation for the search Bit and $\overline{Bit}$ values encoded in all three wavelengths ($\lambda_1$, $\lambda_2$ and $\lambda_3$) and the target Values '0', '1' and 'X' indicatively encoded at wavelengths ($\lambda_1$, $\lambda_2$ and $\lambda_3$) respectively. Figure 5b illustrates their respective eye diagrams. The green highlighted area focuses on the case of the search value of logical "0", meaning the Bit value is "0" and the $\overline{Bit}$ is "1". As observed, when the search value is compared with the target symbol "0", the HD circuit output is zero (HD = 0), meaning that a logical "match" state is acquired. Conversely, when the target symbol is "1", the HD = 1 indicating a "mismatch". For TCAM operation, when the target symbol is the wildcard "X", the output power remains always at zero level, signifying a match with any incoming search value. Similarly, Fig. 5c, d present the respective time traces and eye diagrams, when the WDM-enabled processor operates at 50 Gb/s, revealing successful HD/TCAM functionality. In view of

evaluating the performance at 20 and 50 Gb/s, Fig. 5e, f depict the acquired Q factor values for both the actual and supplementary input (Bit and $\overline{Bit}$) along with the results when targets '1' and '0' are sequentially encoded to different available wavelength cells. Similarly to the single-$\lambda$ 4 × 4 Xbar layout, the 50 Gb/s cases exhibit a performance degradation compared to the 20 Gb/s, stemming from the bandwidth limitation of the deployed electro-optic components.

## Discussion

Following the experimental validation of photonic Xbar-based CAM and HD implementations at up to 50 Gb/s and 3λ parallel layouts, we proceed by developing an analytical framework capable of: (i) assessing the theoretical error rate performance of Xbar-based HD and CAM-layouts for different scales, operational rates and injected laser powers (ii) quantifying the required single-$\lambda$ laser power for different HD, CAM and WDM HD and CAM scales and operational rates and (iii) projecting the achieved energy efficiency of the different targeted HD and CAM single-$\lambda$ and WDM layouts. A detailed breakdown of the opto-electronic noise sources and different properties of the constituent photonic and electronic components incorporated in the analytical framework can be found in Supplementary Material Section 3.

Figure 6a indicatively showcases the simulation framework's projected error rates vs input laser power for HD and CAM operation, when targeting a single-$\lambda$ 32×32 Xbar based layout, equivalent to a search word bit length of 16 (N/2) and a stored word capacity of 32 (N). The simulation results highlight: (a) the increased input laser power requirement when targeting high-rate operations, originating from the increased noise-bandwidth and as such noise profile of the photonic layout (b) decreased input laser power requirements when targeting CAM-only operations i.e. the specific case of HD = 0. This decrease is attributed to the binomial statistical distribution of the power levels arising at the output of the HD photonic processor, originating from the actual and complementary nature of the input representation and the analog XOR operation between the input and target vector. (c) A different error-rate response when CAM-only operations are targeted, as highlighted by the black rectangle in Fig. 6a. This is due to the fact that the noise percentage characterizing the error-rate response of each signal level decreases more slowly, from the highest to the lowest power signal level, as the laser power increases. Additionally, since each signal level is associated with unequal probabilities of occurrence at the output of the system, this effect is reflected in the corresponding graphs (detailed correlation is described in Supplementary Section 3). By setting an operational threshold comparable to the performance of state-of-the-art electronic CAM implementations (i.e. Error Rate of $10^{-3}$)[35–37],

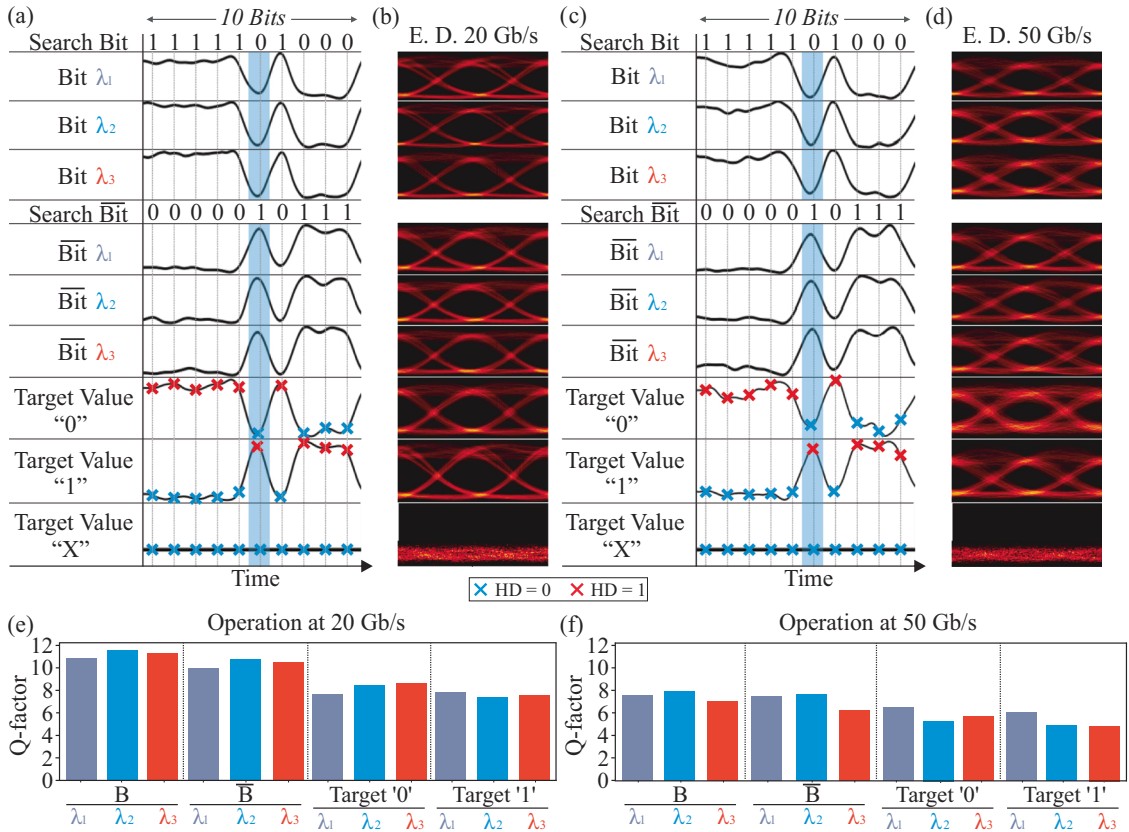

**Fig. 5 | Experimental results and Q factor values for the validation of the WDM comparison operation, with the use of 3 different wavelengths. a** Synchronized time traces (50 ps/div) of 10 optically encoded search Bit and $\overline{\text{Bit}}$ values at three different wavelengths (1548.3 nm, 1558 nm and 1554.9 nm) and the comparison traces assigned to the symbol values of "0", "1", and "X", at 20 Gb/s. **b** Eye diagrams for the respective cases. **c** Corresponding time traces (20 ps/div). **d** Eye diagrams at 50 Gb/s. The traces and eye diagrams for the target values '0', '1' and 'X' are shown as indicative examples at $\lambda_1$. **e** Q factor diagrams for different search bit wavelengths and symbol cases, for 20 Gb/s and **f** 50 Gb/s. All possible combinations of symbol values and wavelengths are depicted at both 20 and 50 Gb/s operation, highlighting the broadband capabilities of the deployed computing cells. WDM wavelength division multiplexing.

we can derive the required single-λ input laser power for different scales and operational rates.

Figure 6b illustrates the required single-λ and 4-λ laser power for the HD, CAM architectures when targeting Xbar layouts with a search word bit length in the range of [2–64] and a stored word capacity range of [4–128]. These layouts correspond to Xbar scales of 4 × 4 to 128 × 128 for the single λ case, and 4 × 1 to 128 × 32 for the 4-λ case, as the 4λ WDM design effectively acts as a capacity enhancement factor of λ = 4. The derived results showcase that WDM layouts, using 4 λ, for both CAM and HD designs, can offer significant insertion loss benefits as compared to the single-λ case (for more details see Supplementary Section 4), that can reach up to 20 dB when targeting 64-bit long search word and 128 stored word capacity for CAM operation at 20 Gb/s. Taking into account, that the currently maximum achievable power of an integrated laser source reaches ~22 dBm[41], the analysis also reveals that WDM designs can also significantly extend the currently achievable CAM designs to up to 64-bit long search words. Finally, Fig. 6c puts in juxtaposition the achieved energy efficiency of single-λ and WDM CAM and HD layouts, taking into account the consumption of the constituent electro-optical components and the increased number of laser sources required for WDM layouts (For details see Supplementary Section 4). In Xbar scales where the required laser power surpasses the current practical limit i.e. >22 dBm, energy efficiency deteriorates because the total power consumption increases substantially and gets dominated by the laser's power requirements. The results also showcase the power consumption reduction properties of WDM layouts, that get even

more significant as the targeted scale increases. This highlights the capabilities of photonic WDM CAM layouts to achieve down to 400 fJ/bit and 200 fJ/bit energy efficiencies, while reaching operating speeds of 20 Gb/s and 50 Gb/s, up to 20 times higher than state-of-the-art electronic counterparts[2,42–44].

## Methods

### Experimental setup of single-wavelength 4 × 4 crossbar-based hamming distance processor

A tunable laser source (Santec TSL 550) is employed to generate a CW signal at the wavelength of 1563 nm. The operational wavelength has been appropriately selected to comply with the response of the grating couplers (GCs) and the EAMs, providing an optimal extinction ratio (ER), as well as insertion losses (ILs). The optical signal, is propagated into the 4 × 4 Xbar-based comparison structure within the PIC through an angled multi-port fiber array unit, retaining a constant TE polarization mode. The GCs of the integrated structure were found to have an IL of around 3.5 dB, which is independent from the Xbar scale. After entering the structure, the signal is split (1 × 4 splitting stage) and gets injected into the four-search word vector EAMs, with an electro-optical (EO) bandwidth of 56 GHz. Each EAM is driven by a waveform generator (AWG Keysight M8194A), generating non-return-to-zero (NRZ) pseudo random bit sequences (PRBS[7]) at 50 Gb/s, in order to emulate the incoming search vector information cases. The AWG generates the electrical signals with the channels having a power raging between [270–380 mV], for all the

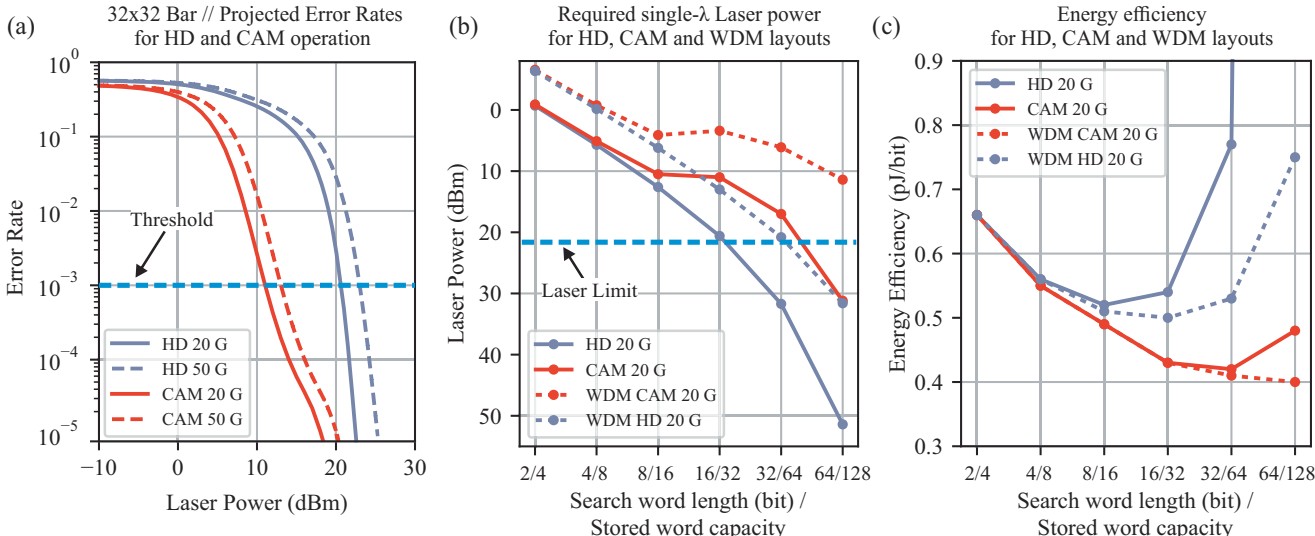

**Fig. 6 | Theoretical framework for higher scale HD and CAM architectures for the operational data rates of 20 and 50 Gb/s, considering single and multi-wavelength layouts. a** Projected Error Rates for a 32 × 32 crossbar implementation at 20 and 50 Gb/s operation. The CAM operation refers to the specific HD case HD = 0, and as such is correlated to the MER metric, while the HD case is correlated to the SER metric. The operational threshold is set based on state-of-the-art electronic counterparts. **b** Required single-λ and WDM laser power for HD and CAM layouts. The horizontal axis is expressed in search word length (bit)/stored word capacity, in order to provide similar capacity points for single and multi-λ layout comparison. The laser limit refers to the currently achievable maximum power in integrated photonics. **c** Energy efficiency for single-λ and WDM HD and CAM layouts at 20 Gb/s. The respective results for 50 Gb/s operation can be found in Supplementary Section 4. CAM content addressable memory, HD hamming distance, MER match error rate, SER symbol error rate, WDM wavelength division multiplexing.

different cases of the 20 and 50 Gb/s, which were further amplified using RF amplifiers (SHF S804B), reaching the required -1.9 Vpp to drive the EAM modules. A pair of EAMs was exploited to encode the information of a single search bit as a pair of complementary data, and as such the first pair of EAMs imprints the $Bit_1$ and $\overline{Bit}_1$, while the second pair the $Bit_2$ and $\overline{Bit}_2$ values. The encoded search bits of the second EAM pair were time-shifted during the electrical signal generation stage relative to the first EAM pair, enabling the acquisition of all possible 2-bit combinations. The search vector values are propagated into the device, where the EAMs of the array are statically assigned to encode all the different binary and ternary vector values, being configured with the use of a multi-channel DC control plane. The DC plane incorporates 8 different programmable 2-channel DC power supplies (Rigol DP831 LXI) that apply the necessary voltage values to every EAM module. Additional information about the specific driving voltage values can be found in the Supplementary Section 1 of the Supplementary Material. Each optical signal at every Xbar output column, representing the logical comparison between the incoming search and statically assigned vectors, exits the optical chip via the same fiber array. It is then amplified by an EDFA, filtered through an optical band-pass filter (OBPF) with a 0.8 nm bandwidth, and directed to a 70 GHz photodiode (Finisar XPDV3120) before being recorded by a 75 GHz sampling oscilloscope (Keysight DCA-X N1000A). Moreover, the EAMs that were responsible for the generation of the search vector values have been reversely biased at voltages ranging between [−1.6, −1.4 V] for 20 Gb/s, while the biasing values were ranging between [−1.6, −1.3] for the 50 Gb/s cases. In both operational cases the EAMs of the comparison cells were biased with either 0 or −4V. Furthermore, a 4th-order butterworth software-based filter was implemented via the OSC module and applied to the captured signals, with a manually adjusted 3 dB bandwidth for the 20 and 50 GHz, respectively, in order to mitigate the excess noise bandwidth response of the transmitting channel. Pattern correlation and error rate calculations were performed off-line.

## Experimental setup of WDM comparison cell

Three TLSs are deployed to generate three CW signal beams at the wavelengths of, $\lambda_1 = 1548.4$ nm, $\lambda_2 = 1558$ nm and $\lambda_3 = 1554.9$ nm. The three signals are then multiplexed via a multiplexer (Mux) module (details about the design of (de)mux are given in Supplementary Material Section 2), creating a WDM stream. The combined signal stream is amplified by an EDFA prior entering the chip, through an angled multi-fiber array. A constant TE mode is ensured for the signals upon entering to the respective GCs, via in-line polarization controllers during CW signal generation. The GCs are calculated to have ~4 dB of ILs. Within the optical chip, the WDM signal is split and directed to two input EAMs of the SVEU, each with an EO bandwidth of 56 GHz. Each input EAM was driven by an AWG (Keysight M8194A) to generate, as in the single wavelength case, 50 Gb/s PRBS[7] data sequences for the Bit and $\overline{Bit}$ complementary search value information. The generated electrical signals have amplitudes ranging between [260−370 mV], at the 20 Gb/s and 50 Gb/s cases, being amplified by RF amplifiers (SHF S804B), reaching the necessary voltage of ~1.9 Vpp to drive the EAMs. After the optical generation of the Bit and $\overline{Bit}$ signals, the WDM stream is propagated to the De-Mux components of each arm of the "outer" MZI structure. Every branch contains a demultiplexer (De-Mux$_1$ and De-Mux$_2$), which filters the incoming WDM stream up to three different paths, each containing a statically configured EAM and a PS, driven by the same DC control plane mentioned above (single-wavelength 4 × 4 Xbar-based HD processor). All possible combinations between the wavelengths ($\lambda_1 = 1548.4$ nm, $\lambda_2 = 1558$ nm, $\lambda_3 = 1554.9$ nm) and symbol values ("0", "1", "X") have been investigated. The optical signals are then multiplexed by the corresponding multiplexers (Mux$_1$ and Mux$_2$) and the two recombined WDM streams are constructively interfered before exiting the Xbar through a single output port. The output of the photonic crossbar is filtered to each wavelength-constituent and was amplified via an EDFA, followed by a BPF with 0.8 nm bandwidth prior entering a 70 GHz PD (Finisar XPDV3120) and being recorded by a 75 GHz OSC (Keysight DCA-X N1000A). Additionally, the EAMs used for the search bit's information have been reversely biased at voltages between [−1.5, −1.2 V] for all the cases of the 20 and 50 Gb/s, while the

EAMs that were used for the statically configured cells have been biased at 0 or −3V. As before, a software-based 4th-order butterworth filter is implemented and applied via the OSC upon receiving the signals, with manually adjusting a 3 dB bandwidth of 20 or 50 GHz in order to mitigate the excess noise response of the PD. Pattern correlation and Q-factor calculations were performed off-line.

## Data availability
The data cases that support the experimental results and findings of the study, as well as all the validation methods and procedures for the additional analysis performed are provided in Supplementary information files. In case of need for additional information and material of the study the authors can provide them upon request. Source data are provided with this paper.

## Code availability
All software and source code files, as well as documentation and instructions for the analysis and validation of the experimental and theoretical findings of the study are provided in Supplementary information files and documents. Any additional specifications regarding the methods that have been followed can be provided upon request by the authors.

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

## Acknowledgements

This work was supported by the European Commission (EC) through the H2020 projects GATEPOST (101120938) and OCTAPUS (101070009).

## Author contributions

T. Moschos, C. Pappas, and N. Pleros conceived the experiment. T. Moschos, C. Pappas, S. Kovaios, I. Roumpos and A. Prapas deployed the experimental testbeds for the 4×4 and the WDM crossbar prototypes, performed the experiments and processed the experimental results. T. Moschos, A. Tsakyridis, M. Moralis-Pegios, C. Vagionas and N. Pleros contributed to the preparation of the manuscript's material and performed the simulation analysis provided. Y. London, T. Van Vaerenbergh and B. Tossoun contributed to the discussion on the original idea and on the organization of the results and the manuscript. All authors discussed the results and contributed in the writing process of the manuscript.

## Competing interests

The authors declare no competing interests.
