## [Transparent Peer review file · Nature Communications]

Nonlinear Optical Vector Processing using Linear Silicon Photonic Circuits for 50 Gb/s Memory and String Similarity functions

Corresponding Author: Mr Theodoros Moschos

Version 0:

Reviewer comments:

Reviewer #1

(Remarks to the Author)

This study marks the first time that nonlinear vector processing based on linear optical circuits has been realized in silicon photonics, offering a novel technological approach for fields such as pattern matching and error correction, and has made certain progress. Overall, this paper is highly innovative and holds promising applications in the field of photonic computing. However, there are some issues with the paper that require further improvement. It is recommended that the authors carefully address the following issues in the revised manuscript and make major revisions to the content of the paper.

1. The paper claims this is the first time nonlinear vector processing has been achieved in silicon photonics, but a 2023 study in Nanophotonics [27] also explored a similar direction. Could you further compare the differences between the methods in this paper and those in references [27-28], especially regarding the error rates of $\sim 10^{-3}$ for HD calculations and CAM operations at 50 Gb/s achieved in this paper, and how they differ from the performance in references [27-28]?
2. The experiment used a 4×4 crossbar structure, but the theoretical simulation part was extended to a 32×32 scale. The paper mentions an insertion loss of about 3.5 dB. Has the change in insertion loss and crosstalk in larger-scale structures (such as 32×32) been evaluated?
3. The paper mentions the use of a 56 GHz SiGe EAM but does not explain why this parameter was chosen. Is the 56 GHz bandwidth crucial for 50 Gb/s operation, especially considering the error rate of $\sim 10^{-3}$ observed in the experiments in this paper? Have the performance of other modulators (such as MRR) been considered for comparison?
4. In Figure 3, the eye diagram opening at 50 Gb/s is significantly smaller than that at 20 Gb/s. The paper states this is mainly due to the bandwidth limitations of the electro-optical components. Has the specific impact of noise sources (such as thermal noise and crosstalk) on the bit error rate been quantitatively analyzed?
5. The paper mentions that the error rate of $\sim 10^{-3}$ is comparable to that of electronic devices [31-33], but it does not clearly state the specific advantages of the photonic scheme in terms of power consumption and delay. Could you supplement the energy efficiency comparison data with electronic CAMs, especially regarding the energy efficiency of 400 fJ/bit and 200 fJ/bit achieved by the WDM layout mentioned in this paper, and what are the specific advantages compared to the pJ/bit of electronic devices?
6. The WDM scheme increases capacity through wavelength multiplexing, but the experiment only verified the performance of 3 wavelengths in parallel. How is the performance of the 3-wavelength parallel operation mentioned in the paper? Has the performance with more wavelengths (such as 4λ or 8λ) been tested, especially the impact of multi-wavelength crosstalk on matching accuracy?
7. In the current WDM scheme, the lasers and multiplexing/demultiplexing modules have not yet been fully integrated into the on-chip system. How is the performance of the 3-wavelength parallel operation mentioned in the paper? If full integration is achieved in the future, thermal drift may cause wavelength misalignment. Are there plans to introduce temperature control solutions (such as on-chip thermal phase shifters) or dynamic wavelength calibration technologies (such as feedback control circuits) to mitigate thermal stability?

Reviewer #2

(Remarks to the Author)

In this work, the authors demonstrated high-speed Hamming distance (HD) calculations and content addressable memory (CAM) bank operations using linear silicon photonic circuits. The first silicon photonic circuit was initially developed for optical matrix multiplications and reported in their previous work [Ref. 4]. The second circuit uses wavelength-division multiplexing (WDM) to enable parallel processing, which also originates from their previous works [Refs. 34 & 35]. The use of these linear photonic circuits for HD calculations and CAM operations is novel. However, the potential impact of performing these operations in the optical domain is not very clear, given the additional complexity introduced by the photonic circuits. Before I can make a recommendation, the authors should consider the following concerns and comments.

Major concerns:

1. The silicon photonic circuits in this work were initially developed for matrix multiplications, which have significant applications in optical neural networks. However, in this manuscript, the authors focus on HD calculations and CAM bank operations, which, while important for specific applications such as pattern matching, may not have the same broad technological impact. The scope of these operations is relatively narrow compared to the extensive influence of matrix multiplications in neural network accelerators. As a result, the work may not achieve the level of general interest expected for publication in a high-impact journal such as Nature Communications.
2. A comprehensive comparison between the demonstrated photonic circuits and previous digital and analog electronic circuits is lacking. The authors claim that their photonic circuit outperforms "state-of-the-art CAM speed performance by > 2.5x." However, this evaluation focuses solely on speed and does not consider other critical metrics. Factors such as chip size, system complexity, and overall energy efficiency are equally important when assessing the practicality of a hardware architecture. Without addressing these aspects, it is difficult to fully understand the trade-offs involved in using photonic circuits for these operations. In particular, the scalability of this scheme could be an issue. To process an N-bit binary vector, this scheme requires 2N modulators to generate the actual and complementary values. This can significantly limit the scalability of this scheme in terms of chip size.
3. The operation principle in Fig. 1(d) is not clearly explained. The authors mentioned "...adding coherently the resulting optical fields". However, Fig. 1(d) only depicts intensity modulators. Does the right part of Fig. 1(d) represent a cascade stage of 1-by-2 optical splitters? Phase shifters are necessary for coherent light summation, which are not shown in this figure. While the supplementary material does provide more details on the operation principle, it should be sufficiently clear within the main text and figures.
4. What is the extinction ratio of the SiGe electro-absorption modulator (EAM)? Could an improvement in the extinction ratio further reduce the error rates in Figs. 3(e-f)?
5. Page 4, line 116. Please provide more details on the multi-channel DC control plane.

Minor comments:

6. The current title may be too broad. "Nonlinear optical vector processing" does not accurately summarize the HD calculations and CAM bank operations.
7. The authors should try to use high-resolution images of their circuits. The current images have poor resolutions, particularly in Fig. 2(a) and Fig. 4(b). No effective information is available from the current figures.
8. Page 1, line 21. Please provide the definition of "WDM".
9. Page 4, line 123. Please provide the definition of "ML".
10. Please use the correct multiplication sign "×" instead of the letter "x" throughout the manuscript.

Reviewer #3

(Remarks to the Author)

The manuscript presents a novel approach to nonlinear optical vector processing using linear SiPho circuits, achieving 50 Gb/s Hamming Distance (HD) and CAM operations. However, several limitations in experimental validation, scalability analysis, and technical comparisons need addressing to strengthen the claims and ensure reproducibility. I will suggest a comment of Major Revision.

1. Expand experimental validation to larger-scale architectures.
To conduct small-scale extended tests (e.g., 8×8 or 16×16 X-bar) to verify theoretical scalability models. To discuss manufacturing challenges (e.g., crosstalk, cumulative losses) for large-scale integration.
2. Enhance error rate analysis with noise source quantification and energy efficiency measurements.
To quantify or evaluate the contributions of thermal noise, crosstalk, and nonlinear effects to SER/MER. To provide measured energy efficiency data (including laser and driver circuitry) for WDM vs. single-λ designs. It seems that the energy efficiency has not consider the optical-electrical conversion power.
3. Update technical comparisons with state-of-the-art electronic CAM/TCAM benchmarks.
To compare against recent CMOS advancements (e.g., 5nm-node TCAMs) in speed, power, and area efficiency.
4. Clarify WDM engineering challenges and mitigation strategies.
To detail wavelength stabilization methods and difficulty (e.g., thermal feedback loops) to counter temperature drift. What is the inter-channel crosstalk suppression (e.g., optimized filter bandwidths, guard bands)?

Version 1:

Reviewer comments:

Reviewer #1

(Remarks to the Author)

From an academic and research point of view, for conceptual proof, small-scale architecture is acceptable. The authors have answered all the questions from the three reviewers, and I believe the answers are accurate. There are no further comments.

(Remarks on code availability)

Reviewer #2

(Remarks to the Author)

Please see the attached document.

(Remarks on code availability)

I thank the authors for addressing some of my concerns. In response to my comment #2, the authors provided “Table 2: State-of-the-art electronic TCAMs” in the revised supplementary document.

Supplementary Table 2: State-of-the-art electronic TCAMs

	Ref.	CMOS Techn. (nm)	Area/cell (μm^2)	Energy/search/bit (fJ)	Freq. (GHz)	Array size
Electronic TCAM	[19]	250	316.2	17.2	0.26	128×32
	[20]	180	53.2	2.82	0.21	128×144
	[22]	130	6.73	5.6	0.20	9.4 Mb
	[23][22]	65	-	-	0.50	-
	[24]	65	6.53	0.2	0.13	32 Kbit
	[25]	65	3.98	0.6	0.50	8 Kbit
	[26]	32	0.84	0.58	1.00	2048×640
	[27]	28	0.6	-	0.40	20 Mbit
	[28]	28	0.304	0.74	0.37	32×64
	[29]	16	1.8	-	1.25	10 Kbit
	[30]	14	2.01	-	1.40	2K×640 bit
	[31]	12	0.945	-	1.50	10 Kbit
	[32]	7	0.248	-	1.60	640 Kbits
	[33]	5	0.287	-	2.00	512×220
	[34]	3	0.201	0.305	2.20	512×220
[35]	2	0.024	0.120	-	128×128	
This Work		SiPho EAM	95,000	32	50	4×4

While it is true that the chip in this work operates at a significantly higher clock frequency (50 GHz) than electronic TCAMs, its energy efficiency (energy/search/bit), array size, and area per cell are significantly worse than those of its electronic counterparts. For example, compared with Ref. 26, which uses an outdated CMOS technology node of 32 nm, the area/cell in this work is 113,095 times larger and the energy/search/bit is 55 times higher; compared with Ref. 34, which uses an advanced CMOS node of 3 nm, the area/cell in this work is 472,637 times larger and the energy/search/bit is 105 times higher.

More importantly, it is nearly impossible that the array size of a photonic circuit can reach the same scale as that of electronic TCAMs, which can easily achieve 512×220. The array size is an important parameter that cannot be ignored in TCAM applications. Lightmatter and Lightelligence have indeed presented large-scale photonic chips for matrix multiplications (128×128 and 64×64, respectively). However, the operating clocks of their photonic chips are 2 GHz and 1GHz, respectively. I am not convinced that each modulator can still be operated at 40 GHz in a practical scenario when the photonic circuit demonstrated in this work is scaled up (for example, to 128×128).

Therefore, while I think this work provides a novel approach, I do not think that the results presented in this work represent a significant advancement compared to electronic TCAMs. Thus, I cannot recommend its publication in Nature Communications.

Aristotle University of Thessaloniki

Department of Informatics
54124 Thessaloniki, Greece

Tel: +30 2310 990588

E-mail moschost@csd.auth.gr

5th July 2025

Dear Reviewers,

We would like to thank the reviewers for their prompt reply and constructive criticism on addressing the respective comments, regarding our submitted manuscript in Nature Communications, entitled “**Nonlinear Optical Vector Processing using Linear Silicon Photonic Circuits for 50 Gb/s Memory and String Similarity functions**”, since we have modified the manuscript’s title based on one the reviewer’s consideration, by T. Moschos, C. Pappas, S. Kovaïos, I. Roumpos, A. Prapas, A. Tsakyridis, M. Moralis-Pegios, C. Vagionas, Y. London, B. Tossoun, T. Van Vaerenbergh and N. Pleros.

With this reply letter you can find attached our respective changes and our thorough replies to the comments provided, along with the respective changes in the overall manuscript material.

We hope that these changes will meet the appropriate requirements and will cover the concerns of the reviewers and thus lead to the consideration of our work being published in the specific scientific journal.

Best regards,
Theodoros Moschos
Department of Informatics
Aristotle University of Thessaloniki

**Revisions on the scientific journal of Nature Communications for the submitted manuscript of
“Nonlinear Optical Vector Processing at 50Gb/s using Linear Silicon Photonic Circuits”-
NCOMMS-25-11055-T**

Reviewer #1

This study marks the first time that nonlinear vector processing based on linear optical circuits has been realized in silicon photonics, offering a novel technological approach for fields such as pattern matching and error correction, and has made certain progress. Overall, this paper is highly innovative and holds promising applications in the field of photonic computing. However, there are some issues with the paper that require further improvement. It is recommended that the authors carefully address the following issues in the revised manuscript and make major revisions to the content of the paper.

Comment #1:

The paper claims this is the first time nonlinear vector processing has been achieved in silicon photonics, but a 2023 study in Nanophotonics [27] also explored a similar direction. Could you further compare the differences between the methods in this paper and those in references [27-28], especially regarding the error rates of $\sim 10^{-3}$ for HD calculations and CAM operations at 50 Gb/s achieved in this paper, and how they differ from the performance in references [27-28]?

Reply:

We would like to thank the reviewer for his/her kind remarks and the constructive comments. In response to the first comment, we would first like to clarify the differences between our current work and the studies cited in references [27] and [28] of the main manuscript.

- **Reference [27]** proposes and analyzes WDM and TDM optical TCAM architectures using silicon photonic microring resonators as the primary comparison units, presenting:
 - Just simulation studies for 10Gb/s search operations
 - and experimental validation at an operational rate of only 4 Gbps.
- **Reference [28]** presents the first experimental demonstration of a 20 Gbps optical TCAM cell using a crossbar configuration that utilizes solely the space dimension.

The current work extends well beyond the previous demonstrations along the following items:

- a) **It demonstrates, for the first time, a complete 2-symbol optical word comparison**, while references [27] and [28] demonstrated only single-cell and single-symbol search operations.
- b) **It performs at a record-high operational speed of 50 Gbps, i.e. 12.5x faster** than the search speed reported for microring-based WDM and TDM optical TCAMs reported in [27], **and 2.5x faster** than the single TCAM cell in [28]. To the best of our knowledge, this represents the highest reported data rate for optical CAMs and Hamming distance comparison operations to date.
- c) **It introduces a new architectural scheme that combines both spatial and wavelength dimensions**, providing a highly efficient layout for scaling to multi-bit demonstrations.
- d) It goes beyond conventional CAM cell operation reported in [27] and [28] and **addresses two different non-linear computing processes, i.e. Hamming distance and ‘CAM Match-line’**. We also incorporate an analytical framework to efficiently compute both the match-error rate (MER) and the symbol-error rate (SER) for: (i) the experimentally obtained data, and (ii) a theoretical model of larger-scale crossbar architectures. This model accounts for the noise characteristics associated with multi-level signal scenarios, as thoroughly detailed in the supplementary material. This extended error-rate analysis, along with the use of multiple signal metrics, introduces a new signal processing perspective within our 2-symbol, 4×4 comparison unit—something not addressed in the previous demonstrations of [27] and [28]. The analysis distinguishes between the two error types based on the intended evaluation objective: The calculation of MER quantifies the likelihood of failure in detecting a Match-

Line event, where the input search vector is identical to the stored content (i.e., Hamming distance = 0). The focus here is on accurately distinguishing the ‘0’-signal power level from the noise levels of all other signal states. The calculation of the SER, on the other hand, evaluates the system’s ability to resolve various multi-level signals that arise from differing degrees of mismatch between vectors. It considers the aggregate noise impact across all signal levels, yielding a comprehensive measure of signal integrity. Compared to the referenced works, the present study introduces a more accurate and nuanced error model that incorporates: (i) the binomial distribution behavior of the XOR operation at the comparison output, (ii) noise-dependent variability across the signal levels, and (iii) the architectural implications of scaling the crossbar to higher dimensions.

- e) Finally, our work establishes an error rate threshold of 10^{-3} , which is well-aligned with the standards employed in multi-bit electronic HD and CAM computing cells [1]-[3]. The authors in references [27] and [28] demonstrated 1-bit comparison operations, achieving error rates in the range of $[10^{-5} - 10^{-12}]$ comparable to those achieved in our WDM-based scenario, where 1-bit comparison is also targeted.

In order to make these differences clearer and complying with the reviewer’s comment, we introduce the following changes in our manuscript:

Changes in the text:

In the “Results” section in page 4 in line 103, we have added the following: *“...The functionality of the proposed analog photonic HD processor architecture was assessed via a 4×4 SiPho integrated Xbar layout that is capable of 2-bit vector distance processing, surpassing prior optical CAM implementations [27] [28], which have demonstrated only 1-bit comparison operations”*

In the “Results” section in page 6 in line 153, we have added the following:

“...As can be observed, all error-rates are in the order of 10^{-3} , comparable with the performance achieved by multi-symbol electronic HD and CAM computing cells”

Comment #2:

The experiment used a 4×4 crossbar structure, but the theoretical simulation part was extended to a 32×32 scale. The paper mentions an insertion loss of about 3.5 dB. Has the change in insertion loss and crosstalk in larger-scale structures (such as 32×32) been evaluated?

Reply:

There is probably a misunderstanding regarding the 3.5 dB loss. As is explicitly stated in Section methods, first paragraph, line 240 (“*The GCs of the integrated structure were found to have an IL of around 3.5 dB*”), this value refers to the losses of a standalone grating coupler, which does not relate to the Xbar scale. The insertion loss for various Xbar scales was calculated using the theoretical framework, thoroughly analyzed in [4], as also mentioned in section 3 of the supplementary material. Therefore, we have already evaluated the insertion loss evolution in higher scale crossbar structures.

Figure 1: Average dot product error, induced by the crosstalk of waveguide crossings for various Xbar scales.

To address the reviewer’s comment, we should also take into account the crosstalk induced by the waveguide crossings of the current architecture. Considering the deployment of waveguide crossings presented in the work of reference [5], that demonstrated -55 dB of crosstalk, we performed a simulation analysis, based on this specific performance metric. For the theoretical model we used a Monte Carlo analysis to evaluate the impact of crosstalk and scaling on the performance of our architecture. The study involves varying the size of a square crossbar array, across values of $N = [4, 8, 16, 32, 64, 128]$, while maintaining a fixed crosstalk level of -55 dB. For each crossbar scale, the simulation conducts 100 trials to compute the dot product between the search and target vectors, both initialized with discrete binary values. The dot products are calculated under two conditions: (i) an ideal scenario and (ii) a noisy scenario where crosstalk-induced interference is introduced to the layout. In the case where noise is applied, the dot products are derived by applying a crosstalk factor, corresponding to the -55 dB and additionally, random phase shifts ranging from $[0-2\pi]$ are considered for the incoming search vector’s signals. Following this, the average error between the ideal and noisy dot products is then calculated. Figure 1 illustrates the simulation results showing the average dot product error, induced by the waveguide crosstalk, for different crossbar scales. As shown, the performance degradation due to crosstalk remains negligible even for crossbar scales as high as 128×128 , reaching only $\sim 8 \times 10^{-3} \%$, showcasing that this specific crosstalk parameter does not limit the scalability of the architecture.

Changes in the text: In methods section, page 9, line 246, we have added the following: *“The GCs of the integrated structure were found to have an IL of around 3.5 dB, which is independent from the Xbar scale”*.

In supplementary material, we have added a section “Crosstalk and Extinction Ratio analysis”, where the crossings-induced crosstalk is investigated.

Comment #3:

The paper mentions the use of a 56 GHz SiGe EAM but does not explain why this parameter was chosen. Is the 56 GHz bandwidth crucial for 50 Gb/s operation, especially considering the error rate of $\sim 10^{-3}$ observed in the experiments in this paper? Have the performance of other modulators (such as MRR) been considered for comparison?

Reply:

The deployed SiPho chip was fabricated on imec’s ISIPP50G platform using PDK (process design kit)-ready components. Therefore, the 56 GHz bandwidth of EAMs was not intentionally chosen, but rather inherited from the platform’s standard EAM design. While this high bandwidth offers advantages for high-speed modulation, it is not essential for 50 Gb/s operation. Furthermore, the bandwidth of the deployed electro-optic link in our case is mainly determined by the digital-to-analog converter (DAC), which has an analog bandwidth of 45 GHz, the lowest among all system components. Nevertheless, despite this bandwidth

Figure 2: Projected error rates for a 4x4 Xbar implementation at 50 Gb/s CAM and HD operation under different search vector’s ER conditions. An ER of 6 dB corresponds to microring modulators, while ER ≥ 10 dB corresponds to Mach-Zehnder modulators.

value is lower than the targeted operational rate, our experimental measurements reveal that 50Gb/s performance could still be successfully obtained in the case of NRZ data signal modulation, as shown by Fig. 3(d) and (f) in the main manuscript. Figure 3 (d) illustrates in its first two rows the eye diagrams acquired during 50 Gb/s NRZ modulation of respective EAMs to produce the two bits of the search vector (Bit₁ and Bit₂), showing clearly eye openings. Moreover, the error-rate for these two signals is $<10^{-9}$, as shown in Fig. 3(f), clearly indicating error-free operation.

The $\sim 10^{-3}$ error-rate is observed only during HD calculation and CAM operation, where 3-level instead of binary signals are obtained at the output of each Xbar column. One of the main sources of error-rate degradation in this case is the limited extinction ratio (ER) of the search-vector-encoding EAMs. These EAMs operate at GHz regime exhibiting an ER of ~ 3 dB, while the EAMs responsible for CAM table encoding operate in a static dc regime and have a sufficiently high ER of ~ 8 dB. However, the resulting 3-level signal will be bounded by the maximum and minimum power levels of the search-vector bits, implying that now 3 different power levels have to be accommodated within an ER of 3dB between the maximum and minimum power level. To assess the impact of the search vector's ER on the error rate, we performed a simulation analysis projecting the error rates of a 4x4 Xbar as a function of the received average power for 50 Gb/s HD and CAM operation under different ER conditions. Figure 2 depicts the simulation results, clearly highlighting that higher ER values reduce the receiver's sensitivity requirements. This suggests that adopting a different modulator technology for search vector encoding could enhance overall system performance. For example, microring resonators (MRRs) typically exhibit ER values of 6 dB [6], which, according to Figure 2, corresponds to a 37% reduction in the receiver's required power to achieve an error rate of 10^{-3} . A further reduction of over 50% can be achieved by employing Mach-Zehnder Modulators (MZMs) for search-vector-encoding, which can yield ER values of ≥ 10 dB [7]. However, each modulator technology introduces distinct requirements and limitations. For instance, MRRs are sensitive to temperature variations, while MZMs significantly increase the footprint. These trade-offs should be carefully evaluated when designing a fully integrated system in order to apply the proper mitigation strategies and finally ensure optimal performance.

Changes in the text: In supplementary material, we have added a section "Crosstalk and Extinction Ratio analysis", where the ER limitation is investigated.

Comment #4:

In Figure 3, the eye diagram opening at 50 Gb/s is significantly smaller than that at 20 Gb/s. The paper states this is mainly due to the bandwidth limitations of the electro-optical components. Has the specific impact of noise sources (such as thermal noise and crosstalk) on the bit error rate been quantitatively analysed?

Reply:

We would like to thank the Reviewer for his/her comment. To begin with, we would like to refer the Reviewer to comments #2 and #6, where a quantitative analysis of both coherent and in-coherent crosstalk was performed, revealing negligible signal distortion. Considering the breakdown of the other noise sources and the relationship with the achieved BER, our approach is two-fold:

- The simulations results presented in the discussion section of the manuscript, are derived from an in-house developed optical simulator, that considers shot, thermal and RIN noise as extensively discussed in Supplementary Section part 3.
- A quantitative analysis of the noise contributions in the experimental results presented in Figure 3 of the main manuscript follows.

Our analysis begins by depicting in Figure 3, the optical eye-diagram analysis of the 20 Gbit/s NRZ signal. Specifically, Fig. 3 (a) depicts the optical eye diagram, with the yellow, red and green lines corresponding to the crossing points, the 20%, 80% transitions, and the Y axis eye center, respectively. Figure 3 (b) illustrates the probability density plots at the 10% percentage boundary of the X-axis eye center for the two

Figure 3: (a) Optical Eye Diagram of 20 Gbit/s NRZ signal (b) Horizontal histogram in the 10% boundary of the X-axis center (c) Fitted gaussian distributions characteristics for binary zero and one levels.

binary levels. Fitting a gaussian distribution in each binary symbol class allows the calculation of the mean and standard deviation values of the noise profiles, with the respective results along with other eye-diagram metric illustrated in Fig. 3 (c). Observation of the experimental data leads to some important conclusions:

- The noise profiles at both binary levels follows a gaussian distribution.
- As the noise profile of the ‘1’ binary level is significantly higher than the respective ‘0’ level, the expected noise seems to be dominated by a receiver power dependent phenomenon, suggesting that thermal noise is not the highest noise contributor.

To quantify the contribution of the digital sampling scope thermal noise, we plot in Fig. 4, the probability density plot of a scope measurement, when no input power is injected and a 20 GHz low-pass filter is applied to the sampling scope signal. Fitting a gaussian distribution to the density plot reveals a standard deviation of 0.33 mV, that as expected is significantly lower than the measured 0.58 and 1.1 mV standard deviations at the zero and one binary levels respectively.

Having isolated the thermal noise contribution of the receiver, we experimentally replicated, using DC signals, the opto-electronic component chain employed in our experiments. Specifically, a tunable laser source tuned to the same wavelength as in the experiment, was attenuated to match the power level arising at the chip output (-27 dBm). Following the attenuated signal was injected into an EDFA amplifier, filtered in a 1 nm 3dB bandwidth optical filter and finally injected into a photodiode and sampled in the digital

Figure 4: Probability density plot of the signal at the digital sampling scope, after a 20 GHz low pass filter.

oscilloscope. The injected power at the photodiode was tuned based on the mean values of the binary zero and one levels of the previously acquired eye diagram (i.e. 18.4 and 36.1 mV, respectively).

Figure 5 (a) illustrates the experimental derived probability density plot for the DC measurements for both binary levels 0 and 1. The respective plots are overlaid in Fig. 5 (b) with the optical eye diagram experimental derived probability density plots, revealing perfect matching. As such, the increased noise profiles at the optical eye diagram measurements, as compared to the scope's thermal noise, is attributed to the EDFA originating noise, with the largest contribution attributed to the beating of the EDFA's ASE with the signal power, resulting in a receiver power dependent noise profile [8].

Figure 5 (a) Probability density plot of DC measurement of noise profile at binary level '0' and '1' at 20 Gb/s (b) Overlaid DC and RF noise profiles

The previous analysis is repeated for 50 Gb/s NRZ signals, with Fig. 6, 7 and 8 illustrating the thermal noise after a 50 GHz low pass filter, the optical eye diagram and respective metrics and the overlaid DC and RF density plots, respectively. It is evident that the noise profiles at 50 Gb/s operations is increased due to higher thermal and signal-to-ASE beating noise at 50 GHz. However, the overlaid plots showcase almost perfect matching, distorted only slightly at the '1' binary level, due to the eye skew originating from the limited frequency response.

Figure 6: (a) Optical Eye Diagram of 50 Gbit/s NRZ signal (b) Horizontal histogram in the 10% boundary of the X-axis center (c) Fitted gaussian distributions characteristics for binary zero and one levels.

Figure 7: Probability density plot of the signal at the digital sampling scope, after a 50 GHz low pass filter.

Figure 8 (a) Probability density plot of DC measurement of noise profile at binary level ‘0’ and ‘1’ at 50 Gb/s(b) Overlaid DC and RF noise profiles.

This analysis leads to the following conclusions, considering the eye closure between 20 and 50 Gb/s operations

- While the electro-optic frequency response is indeed limited, its effect on the eye opening is significantly minimized through a pre-emphasis procedure. However, the pre-emphasis procedure results in a reduction in the achieved ER, from 2.9 dB to 2.1 dB, when transitioning from 20 to 50 Gb/s operation.
- The increased noise profile at higher operating rates and as such operating bandwidths, is also a significant contributor in the reduction of the achieved eye opening.

Quantifying these distortions (ER and increased noise profile) in terms of theoretically achieved Bit Error Rate, can be performed through the use of the optical eye Q-factor :

$$Q = \frac{P_1 - P_0}{\sigma_1 + \sigma_0} = \frac{P_1 - P_1/ER}{\sigma_1 + \sigma_0} = P_1 * \frac{1 - \frac{1}{ER}}{\sigma_1 + \sigma_0}$$

With P1, P0, σ1 and σ0 corresponding to the mean power levels and noise standard deviation of the signal at binary logic levels ‘1’ and ‘0’ respectively.

Changes in the text: In supplementary material, we have added a section “Experimental noise analysis: Impact of different noise sources”, where the experimental noise is investigated.

Comment #5:

The paper mentions that the error rate of $\sim 10^{-3}$ is comparable to that of electronic devices [31-33], but it does not clearly state the specific advantages of the photonic scheme in terms of power consumption and delay. Could you supplement the energy efficiency comparison data with electronic CAMs, especially regarding the energy efficiency of 400 fJ/bit and 200 fJ/bit achieved by the WDM layout mentioned in this paper, and what are the specific advantages compared to the pJ/bit of electronic devices?

Reply:

Ternary features used in vector comparisons, similarity matching as well as look-up tables require developing hardware CAMs/TCAMs, that comprise two storage cells, one for the actual data-bit and one for the ternary state. While TCAMs offer simplicity and high performance, they come at the cost of

increased power consumption and complex interconnect circuitry. This circuitry broadcasts input data across all TCAM match-lines to enable fast parallel comparisons, with the final result collected at the TCAM table output as a single Matchline signal. The first practical integrated electronic CAM was introduced by Koo in 1970 [9], yet emerged in the memory market in mid-90s and flourished after 2000, owing to the established Von-Neumann architectures, favouring enhanced search-performance and memory-intensive operations when compared against conventional electronic RAMs. Since then, TCAMs followed a rapid progress as summarized in the Table 1 that benchmarks various electronics TCAMs optimized for high-performance at various CMOS nodes from 250 nm down to even 2nm.

- Early T-CAMs targeted combining the performance of NOR gates with the power efficiency of NAND gates by activating only a few MLs using NAND cells and NOR cells for the rest, as e.g. the device built on 250 nm CMOS node at 260 MHz [10]. Similarly, a T-CAM at a 180 nm CMOS node achieved a maximum frequency of 210 MHz [11], while devices on 130 nm CMOS exhibited 200 MHz [12], with area and energy consumptions quickly reaching values of 6.73 μm^2 and 5.6 fJ/bit respectively.
- When using CMOS lines of 65 nm, electronic T-CAMs achieved to deliver frequencies up to 500 MHz, at an energy efficiency value of 0.2 fJ/bit and footprint of 3.98 μm^2 per cell in 2015 (not simultaneously at the same device) [13]-[15].
- Although T-CAM developed at CMOS nodes between 32nm and 12nm achieved frequencies in the order of 1GHz to 1.5 GHz [16]-[21], the energy efficiency values were still not improved (values >0.2 fJ/bit are reported), resulting only in 4x increased footprint efficiency of 0.945 μm^2 per cell.

Table 1: State-of-the-art electronic T-CAMs

	Ref.	CMOS Techn. (nm)	Area/cell (μm^2)	Energy/sear ch/bit (fJ)	Freq. (GHz)	Array size
Electronic TCAM	[9]	250	316.2	17.2	0.26	128×32
	[10]	180	53.2	2.82	0.21	128×144
	[12]	130	6.73	5.6	0.20	9.4 Mb
	[13]	65	-	-	0.50	-
	[14]	65	6.53	0.2	0.13	32 Kbit
	[15]	65	3.98	0.6	0.50	8 Kbit
	[16]	32	0.84	0.58	1.00	2048×640
	[17]	28	0.6	-	0.40	20 Mbit
	[18]	28	0.304	0.74	0.37	32×64
	[19]	16	1.8		1.25	10 Kbit
	[20]	14	2.01	-	1.40	2K×640 bit
	[21]	12	0.945	-	1.50	10 Kbit
	[22]	7	0.248	-	1.60	640 Kbits
	[23]	5	0.287	-	2.00	512×220
	[24]	3	0.201	0.305	2.20	512×220
[25]	2	0.024	0.120	-	128×128	
This Work	SiPho EAM	95,000	32	50	4×4	

- More recently, shifting to the most advanced CMOS nodes of lower than 7 nm [22]-[25], energy efficiency values were not significantly improved, revealing a plateau of around 0.12 fJ/bit, bounded by the nature of the underlying electrical interconnect.

All this analysis of the demonstrated electronic state-of-the-art architectures declares that electronic TCAMs have revealed energy efficiencies that can reach even sub-fJ/bit values. However, as shown they are struggling to advance their data-rate performance. Even in the cases of the most advanced electronic nano-sheet layers of NAND-gates at 2nm nodes, operational-speeds seem to be constrained to less than 2.2 GHz, implying that electronic TCAMs are hard-limited within a few GHz and hence struggle to keep pace with the rising line rates of optical systems, downgrading their overall performance.

In this context, photonic CAMs have emerged as a promising alternative computing circuitry, as described also in the main manuscript. The presented 4x4 and WDM TCAM prototypes reveal a limitation in terms of footprint and energy capabilities being characterized by the trade-offs of the current photonic integration, compared to the electronic state of the art works. In the experimental case of the 4x4 layout the required energy for a search operation is approximated to be ~ 32 fJ/bit, occupying an area of 0.095 mm^2 (included in the respective state-of-the-art table 1), with the WDM alternative requiring ~ 38 fJ/bit, with a relatively higher footprint occupation, due to the multiplexer/demultiplexer modules required for the parallel WDM operation. However, the current prototypes deliver up to a 20x times speed enhancement in both cases, over the electronic performance plateau. These speed credentials along with the utilization of the space and wavelength dimensions show some promising computing capabilities.

Moreover, in the case of the WDM implementation, our theoretical analysis highlights the advantage of our SDM+WDM configuration when scaling to higher dimensional crossbar arrays, in terms of energy requirements. For higher scales such as 32-bit configuration and above the energy efficiency for the HD operation remains below 800 fJ/bit in all cases, while the respective CAM operations reveals efficiency values of 400 fJ/bit and 200 fJ/bit respectively, for 20 Gb/s and 50 Gb/s. The WDM implementation can potentially allow minimized laser consumption compared to the single λ layout when scaling to higher crossbar dimensions, showcasing promising energy efficiency characteristics. This analysis shows that a possible scale-up approach for multi-comparison high speed operations can maintain reasonable footprint requirements and energy efficiency metrics. Detailed explanation about the laser consumption reduction can be found in the Supplementary material Section 4.

Addressing the delay aspect, based on the comment of the reviewer we assume that by “delay” he/she refers to the latency time, required to perform a comparison operation in the given architecture. Our experimental work is focused on the modulation operational rate and does not take into consideration any explicit time delay measurements. It needs to be mentioned that photonic integrated circuits offer inherently lower delay metrics compared to their electronic counterparts, especially in the cases of interconnect-dominated and parallel-processing applications. In cases of dielectric media, such as silicon waveguides that we utilise in our photonic circuits, light’s high group velocity, estimated to be $\sim 2 \times 10^8 \text{ m/s}$, reduces significantly the propagation time of the photonic signal, through the respective photonic circuits. Additionally, the presented photonic architectures are characterised by passive, interference-based computation, without requiring sequential logic/switching circuits, that pose additional delay times in the respective CMOS-based electronic schemes. As such the overall system delay in our approach is mostly determined by the optical path routing and coupling efficiencies, with the only limiting factor being the modulator limitation characteristics, which in our case operates at the GHz regime. Hence, we can reach to the conclusion that the signal delay in our photonic implementation can be estimated to be at the ps time regime, being comparable with the electronic alternatives.

Changes in the text: The table with state-of-the-art electronic T-CAMs has been added to the section “State-of-the-art electronic CAM demonstrations” of the supplementary material.

Comment #6:

The WDM scheme increases capacity through wavelength multiplexing, but the experiment only verified the performance of 3 wavelengths in parallel. How is the performance of the 3-wavelength parallel operation mentioned in the paper? Has the performance with more wavelengths (such as 4λ or 8λ) been tested, especially the impact of multi-wavelength crosstalk on matching accuracy?

Reply:

The performance of a 3-wavelength parallel operation has been experimentally verified in our implementation and is thoroughly described in the main manuscript in the section: “WDM-Enabled Comparison Cell Architecture for Scalable HD Processor Layouts”. The proposed WDM-enabled comparison cell facilitates the parallel execution of HD operations within a single output, where the number of wavelengths represents the level of parallelism. Our photonic integrated chip incorporates (de)multiplexers originally designed to support 4 different wavelengths with a channel spacing of ~ 3 nm. However, due to fabrication issues, only 3 wavelength channels were fully functional, while the fourth exhibited significant insertion losses. Therefore, we successfully demonstrated parallel operation of 3 wavelengths at up to 50 Gbps, with the experimental results summarized in Figure 5 of the main manuscript.

We agree with the reviewer’s concern that optical crosstalk might affect the performance of the WDM crossbar layout, particularly as the number of wavelength channels increases and/or the channel isolation performance of the underlying photonic components decreases. In order to quantify the effects of the optical crosstalk in a k number of wavelength channels WDM-crossbar, we begin by depicting in Figure 9 (a) the equivalent circuit layout of a single column of the WDM layout. We also assume a crosstalk value of A dB and symmetric channel operation. Figure 9 (b) illustrates the optical power evolution of the multiwavelength signals traversing the crossbar layout for each of the k wavelength channels. By denoting λ_1 as the signal of interest and identifying all the other wavelength channels as crosstalk inducing aggressors, we conclude to the following:

- Channel 1 crosstalk power:
 - $P_{xtalk1} = (k - 1) * (P_{input} * 3 * MUX_{xtalk})$ (1)

where P_{input} is the optical power for each wavelength channel at the input of the first De-MUX, while MUX_{xtalk} is the isolation of each of the WDM components in linear form and the factor 3 indicates the channel spacing.

- Channel 2-k crosstalk power:

$$P_{xtalk2} = (P_{input} * 2 * MUX_{xtalk}) + (P_{input} * MUX_{xtalk}) + (k - 2) * (P_{input} * 3 * MUX_{xtalk}),$$
 (2)

with the first component being the intra-band crosstalk at λ_1 , the second component originating from $\lambda_{2(k=2)}$ and the $(k-2)$ components corresponding to the crosstalk of all channels to the signal bearing signal.

Adding up these contributions we conclude to:

$$P_{xtalk} = P_{xtalk1} + (k - 1) * P_{xtalk2} + (k - 1) * (P_{input} - 3 * MUX_{xtalk}) + k * \{(P_{input} - 2 * MUX_{xtalk}) + (P_{input} - MUX_{xtalk}) + (k - 2) * (P_{input} - 3 * MUX_{xtalk})\}. \quad (3)$$

Figure 9: (a) Equivalent circuit layout of a WMD-Xbar with k wavelength channels and (b) Schematic illustration of crosstalk aggressors for single column operation.

It should be noted that typical SiPho MUX/DEMUX designs have significant crosstalk only on the case of the adjacent channels, and as such the $(k-1)$ factor on the P_{xtalk2} can be reduced to 2.

Figure 10 (a) and (b) illustrate the relationship between crosstalk to signal power for different multiplexer isolation (crosstalk) and wavelength configurations, taking into consideration the cases of adjacent and non-adjacent crosstalk contributions. Specifically, Figure 10 (a) and (b) illustrate the scenarios where crosstalk is symmetric across all channels and when only adjacent channels contribute to crosstalk power. As can be observed for the case of the adjacent channel crosstalk, a typical performance of -27 dB of crosstalk [26] results in a negligible crosstalk to signal power of 0.004 or 0.4 % and as such we conclude that it would not have any significant effect on the error rates performance.

Changes in the text:

In “Results” section, page 7, line 177, we have added the following:

“It is worth noting that our photonic integrated chip incorporates (de)multiplexers originally designed to support 4 different wavelengths. However, due to fabrication issues, only 3 wavelength channels were fully

Figure 10: Crosstalk to signal power (a) Assuming symmetric crosstalk performance across or multiplexer channels and (b) Assuming only adjacent channel crosstalk.

functional, while the fourth exhibited significant insertion losses. Therefore, we successfully demonstrated parallel operation of 3 wavelengths at up to 50 Gbps”

In supplementary material, we have added a section “Crosstalk and Extinction Ratio analysis”, where the mux/demux-induced crosstalk is investigated.

Comment #7:

In the current WDM scheme, the lasers and multiplexing/demultiplexing modules have not yet been fully integrated into the on-chip system. How is the performance of the 3-wavelength parallel operation mentioned in the paper? If full integration is achieved in the future, thermal drift may cause wavelength misalignment. Are there plans to introduce temperature control solutions (such as on-chip thermal phase shifters) or dynamic wavelength calibration technologies (such as feedback control circuits) to mitigate thermal stability.

Reply:

Indeed, in the current WDM implementation the deployed laser sources, as well as the multiplexer and filter modules have not yet been fully integrated into the photonic chip, as we rely on external components to allow greater flexibility in testing, optimization and proof-of-concept of the WDM CAM operation. Our target for future implementation is to integrate a frequency comb laser device on the transmitter side, eliminating the need for the input multiplexer, while also reducing the footprint and improving energy efficiency. Considering the 3-wavelength operation performance, we would like to kindly refer the Reviewer to comment #6.

Moreover, the reviewer successfully pointed out that thermal drift may lead to wavelength misalignments, potentially hindering the implementation of this layout; therefore, closed-loop temperature control subsystems should be used to ensure thermal stability. In this direction, several approaches have been proposed to lock the resonant wavelength of the ring-based multiplexers, with the dominant method being the employment of energy efficient integrated heaters, [27],[28] as also mentioned by the reviewer. Apart from this solution, one promising approach is to utilize a dithering technique [29] combined with contactless integrated photonic probe (CLIPP) sensors [30], which enable automated wavelength stabilization and thermal drift compensation, by efficiently controlling the working point of each mux/demux in real-time. We believe that this kind of integration will be crucial in realizing a fully embedded WDM CAM system, ultimately improving scalability and deployment feasibility for practical applications.

Reviewer #2

In this work, the authors demonstrated high-speed Hamming distance (HD) calculations and content addressable memory (CAM) bank operations using linear silicon photonic circuits. The first silicon photonic circuit was initially developed for optical matrix multiplications and reported in their previous work [Ref. 4]. The second circuit uses wavelength-division multiplexing (WDM) to enable parallel processing, which also originates from their previous works [Refs. 34 & 35]. The use of these linear photonic circuits for HD calculations and CAM operations is novel. However, the potential impact of performing these operations in the optical domain is not very clear, given the additional complexity introduced by the photonic circuits. Before I can make a recommendation, the authors should consider the following concerns and comments.

Comment #1:

The silicon photonic circuits in this work were initially developed for matrix multiplications, which have significant applications in optical neural networks. However, in this manuscript, the authors focus on HD calculations and CAM bank operations, which, while important for specific applications such as pattern

matching, may not have the same broad technological impact. The scope of these operations is relatively narrow compared to the extensive influence of matrix multiplications in neural network accelerators. As a result, the work may not achieve the level of general interest expected for publication in a high-impact journal such as Nature Communications.

Reply:

We would like to thank the reviewer for his/her insightful comment regarding the scope and impact of our work, as it gives us the opportunity to better clarify the broader significance of our research and its alignment with high-impact applications.

Although the reviewer notes that HD calculations and CAM operations may appear niche, these functionalities are critical for a great range of both traditional and emerging applications such as coding theory, data clustering, **in-memory computing, memory augmented neural networks (MANNs), associative memory systems, and pattern recognition** tasks. These applications are increasingly relevant in areas like communications and networking, computing, bioinformatics, image processing, deep learning and real-time data retrieval, where efficient pattern matching and similarity searches are essential [31]-[34].

For instance, **recent works in the high-impact journal of Nature Communications** [35],[36], have demonstrated CAM-based memory modules for implementing attention mechanisms and associative memory operations in the electronic domain, which are essential for one and few-shot learning tasks. CAM banks allow for rapid content-based retrieval of stored patterns, bypassing the von Neumann bottleneck and enabling analog in-memory computation on high-dimensional vectors. This capability is particularly valuable for tasks such as image classification and pattern recognition, where the ability to quickly identify and retrieve similar patterns from memory is crucial. Beyond MANNs, CAM-based systems also offer transformative potential for accelerating decision trees and random forests, which are top performers in domains requiring interpretability and robust performance with limited training data. Specifically, the authors in [37] have proposed CAM devices for fast look-up table operations, enabling efficient in-memory computation for tree-based model inference.

In addition, such architectures are critical for address look-up (AL) and routing table implementations [38], supporting high-intensity network applications and bandwidth-demanding operations in high-performance computing and data center environments [39],[40]. In these systems, high-speed header processing and packet forwarding are essential for interoperability. However, these functions are often limited by conventional electronic CAMs, which suffer from slow processing speeds due to their electronic structure [35],[37],[41],[42]. Therefore, optical CAM architectures that offer both high-speed operation and energy efficiency represent a valuable advancement in this domain.

HD calculations comprise also critical functions in a wide range of application domains and can certainly not be considered as “not having a broad technological impact”. HD metrics are widely employed in coding theory and error correction codes [43], in information theory [44], in Hamming clustering methods for data clustering [45], as well as in bioinformatics for DNA sequencing [46]. On top of that, our work on HD calculation declares a first successful attempt in the optical domain, using linear circuits for computing a certain distance metric, which may unlock new capabilities for computing a broader range of **distance metrics and similarity measures** that are fundamental to numerous high-impact applications. In particular, our architecture can efficiently perform HD calculations but can be also extended to support other key **non-linear distance metric operations, such as cosine distance and Euclidean distance calculations**, which are essential for data clustering (e.g. nearest neighbor algorithm), classification, and similarity search in machine learning and computer vision tasks.

Therefore, our work not only advances CAM-based photonic computing but also demonstrates the potential of linear silicon photonic circuits to perform a wide array of fundamental linear and non-linear mathematical operations, further broadening its relevance and impact.

Changes in the text:

In “Results” section, line 71, we have added the following:

“HD is a metric widely employed in coding theory and error correction codes [43], in information theory [44], in Hamming clustering methods for data clustering [45], as well as in bioinformatics for DNA sequencing [46], underscoring its wide-reaching impact across computational and scientific disciplines”
We have also added the respective references in the main manuscript.

Comment #2:

A comprehensive comparison between the demonstrated photonic circuits and previous digital and analog electronic circuits is lacking. The authors claim that their photonic circuit outperforms “state-of-the art CAM speed performance by $> 2.5\times$.” However, this evaluation focuses solely on speed and does not consider other critical metrics. Factors such as chip size, system complexity, and overall energy efficiency are equally important when assessing the practicality of a hardware architecture. Without addressing these aspects, it is difficult to fully understand the trade-offs involved in using photonic circuits for these operations. In particular, the scalability of this scheme could be an issue. To process an N-bit binary vector, this scheme requires 2N modulators to generate the actual and complementary values. This can significantly limit the scalability of this scheme in terms of chip size.

Reply:

Regarding the comparison of state-of-the-art electronic CAMs, please refer to the response of Comment #5 of the 1st reviewer. On the other hand, regarding the scalability of the proposed architecture, the reviewer is correct in stating that an N-bit binary processing requires 2N search modulators. However, this is not an oversight but rather a deliberate design choice consistent with the approach used in electronic CAMs e.g. [10]-[15], where both the actual and complementary bit values are generated to enable comparison operations. As such, this requirement is a shared limitation inherent to both electronic and photonic implementations.

Comment #3:

The operation principle in Fig. 1(d) is not clearly explained. The authors mentioned “...adding coherently the resulting optical fields”. However, Fig. 1(d) only depicts intensity modulators. Does the right part of Fig. 1(d) represent a cascade stage of 1-by-2 optical splitters? Phase shifters are necessary for coherent light summation, which are not shown in this figure. While the supplementary material does provide more details on the operation principle, it should be sufficiently clear within the main text and figures.

Reply:

As the reviewers mentions, indeed the detailed explanation of the operational principle of the analog comparison cell has been initially included in the supplementary material of the manuscript. Figure 1 (d) of the main manuscript depicts the main concept of the principle of operation, without illustrating the detailed layout of the architecture along with all the respective building blocks .

Based on the reviewer’s comment we have made the necessary changes in Figure 1 (d) as well as Figure 1 (e) of the main manuscript, in order to present in more detail, the operational principle and the respective building blocks of the comparison circuit. Additionally, changes in the main text have also been performed accordingly.

Changes in the text: The following changes have been performed in the main manuscript, *Section: Hamming Distance using a silicon photonic Xbar PIC.*

- Page 3, Figure 1:

- Page 3, Figure 1 caption:

“Analog photonic HD circuit, based on amplitude modulators and coherent light summation with the use of thermo-optic phase shifters”

- Page 3, line 89:

“(iii) Injecting the encoded X modulated light beams to the Y amplitude modulating elements and adding coherently the resulting optical fields, by utilizing thermo-optic phase shifters that ensure proper phase matching between the two branches. The Comparison Cell highlighted in Fig. 1(d), comprises the basic building block of this comparison operation (details for its functionality are also provided in Supplementary section 1).”

Comment #4:

What is the extinction ratio of the SiGe electro-absorption modulator (EAM)? Could an improvement in the extinction ratio further reduce the error rates in Figs. 3(e-f)?

Reply:

The reviewer can refer to the response we have provided at the *Comment#3* of *Reviewer#1*, regarding the extinction ratio parameter. We have also included the respective analysis in the supplementary material, adding the section “Crosstalk and Extinction Ratio analysis”, where the ER limitation is investigated. Based on our provided analysis it becomes clear that improvement in ER can reduce the receiver’s sensitivity requirements, meaning that the overall performance of our architecture can be improved, taking into consideration the implementation of different modulation technologies, as well as careful design and evaluation approaches of the whole system.

Comment #5:

Page 4, line 116. Please provide more details on the multi-channel DC control plane.

Reply:

Based on the comment of the reviewer, we have added additional information regarding the multi-channel DC control plane that has been used in our experimental evaluations.

Changes in the text: The following changes have performed in the *section : Hamming Distance Calculation using a silicon photonic Xbar PIC*, of the main manuscript.

- Page 4, line 120:
“Both EAMs and TO PSs are controlled by a multi-channel DC control plane, that consists of 16 programmable 2-channel DC power supplies.”
- Page 10, line 257:
“..., being configured with the use of a multi-channel DC control plane. The DC plane incorporates 8 different programmable 2-channel DC power supplies (Rigol DP831 LXI) that apply the necessary voltage values to every EAM module. Additional information about the specific driving voltage values can be found in the section 1 of the Supplementary material.”
- Page 10, line 283:
“..., driven by the same DC control plane mentioned above (single-wavelength 4x4 Xbar-based HD processor).”

Comment #6:

The current title may be too broad. “Nonlinear optical vector processing” does not accurately summarize the HD calculations and CAM bank operations.

Reply:

We have made the following change in the title of the presented manuscript:
“Nonlinear Optical Vector Processing using Linear Silicon Photonic Circuits for 50 Gb/s Memory and String Similarity functions”

Changes in the text: The titles of both the main manuscript and the supplementary material have been updated accordingly.

Comment #7:

The authors should try to use high-resolution images of their circuits. The current images have poor resolutions, particularly in Fig.2(a) and Fig. 4(b). No effective information is available from the current figures.

Reply:

The provided images of the manuscript have been created by a specific graphic software and the current resolution is the highest available that can be exported. Regarding the provided images of Figure 2 (a) and Figure 4 (b), that showcase the integrated photonic structures, have been acquired with the use of a microscope image camera. The current provided figures demonstrate the highest resolution available based on the camera equipment that is currently available in our laboratory. Complying with the comment of the reviewer we have included an inset that indicates the sizing of the layouts of the photonic structures, taking into consideration the dimensions of the integrated components (200 um inset). Based on that, the provided images (Fig.2 (a) and Fig. 4 (b)) have been modified accordingly:

Changes in the text:

- Page 4, Figure 2 (a):

- Page 6, Figure 4 (b):

Comment #8:

Page 1, line 21. Please provide the definition of “WDM”.

Reply:

Based on the comment, WDM stands for *wavelength-division multiplexing*.

Changes in the text: The following changes have performed in the *Abstract* section of the main manuscript.

- Page1, line 21:

“Scalability is enhanced by employing space-wavelength multiplexing schemes via a wavelength-division multiplexing (WDM)-enabled SiPho processor cell, ...”

Comment #9:

Page 4, line 123. Please provide the definition of “ML”.

Reply:

The acronym for the definition ML is referred to *match-line*.

Changes in the text: The following changes have performed in the *Abstract* section of the main manuscript.

- Page 5, line 127:
“The case of $HD=0$ designates an exact vector matching and corresponds to the match-line (ML) operation of a complete CAM memory bank, ...”

Comment #10:

Please use the correct multiplication sign “ \times ” instead of the letter “x” throughout the manuscript.

Reply:

Based on the respective comment, the letter “x” has been replaced with the sign “ \times ”, throughout the whole manuscript.

Changes in the text: The following changes have conducted in the overall manuscript.

- Page 1, line 20:
“... improving state-of-the-art CAM speed performance by $>2.5\times$.”
- Page 2, line 61:
“configured as a 4×4 matrix arrangement and employing 56 GHz bandwidth silicon germanium (SiGe) EAMs as its core computational cells.”
- Page 2, line 64:
“...with matching error-rates of $\sim 10^{-3}$ at operational speeds $2.5\times$ higher than current state-of-the-art CAMs clock speed.”
- Page 4, Figure 2 caption:
“Fig. 2. (a) Microscope photo of the integrated 4×4 crossbar prototype comparison unit. Inset: Single Xbar comparison cell, consisting of two EAMs and two TO PSs. The search vector encoding unit and the 4×4 crossbar, are marked with yellow and orange rectangles, respectively, (b) Experimental testbed for the HD comparison operation at 20 and 50 Gb/s, highlighting the search vector encoding unit (yellow) and the 4×4 Xbar-based HD processor (orange), along with the respective comparison cells (red). A DC control plane is employed for biasing the static EAMs and PSs located in the Xbar layout.”
- Page 6, line 143:
“The performance of the 4×4 vector processor was assessed by calculating the projected symbol error rate (SER)...”
- Page 6, line 169 and 170:
“In order to validate the advantages of the WDM-enabled approach a $3\lambda-2\times 1$ comparison cell was fabricated using the same technological building blocks (i.e. EAM and TO-PS) as in the 4×4 Xbar.”
- Page 6, Figure 4 caption:
“(c) Circuit schematic and experimental setup of the $3\lambda-2\times 1$ HD processor.”
- Page 8, line 186:
“Similarly to the single- λ 4×4 Xbar layout.”
- Page 8, line 198:
“... when targeting a single- λ 32×32 Xbar based layout, ...”
- Page 8, Figure 6 caption:
“Fig.6. (a) Projected Error Rates for a 32×32 Xbar implementation at 20 and 50 Gb/s operation.”
- Page 9, line 214:

“These layouts correspond to Xbar scales of 4×4 to 128×128 for the single λ case, and 4×1 to 128×32 for the $4-\lambda$ case, ...”

- Page 9, line 233:

“Single-wavelength 4×4 Xbar-based HD processor”

- Page 9, line 239:

“After entering the structure, the signal is split (1×4 splitting stage) and gets injected into the four-search word vector EAMs, ...”

Reviewer #3

The manuscript presents a novel approach to nonlinear optical vector processing using linear SiPho circuits, achieving 50 Gb/s Hamming Distance (HD) and CAM operations. However, several limitations in experimental validation, scalability analysis, and technical comparisons need addressing to strengthen the claims and ensure reproducibility. I will suggest a comment of Major Revision.

Comment #1:

Expand experimental validation to larger-scale architectures. To conduct small-scale extended tests (e.g., 8×8 or 16×16 X-bar) to verify theoretical scalability models. To discuss manufacturing challenges (e.g., crosstalk, cumulative losses) for large-scale integration.

Reply:

With all the respect to the reviewer’s comment, the 8×8 and even more the 16×16 Xbar configurations can definitely not be considered as “small-scale extended tests” within the context of a university research laboratory. For instance, testing an 8×8 Xbar layout at 50Gb/s would require 8 independent 50 Gb/s RF channels to be provided by Arbitrary Waveform Generator modules (AWG) similar to the AWG Keysight M8194A that we employed in our experiment. Taking into account that the AWG Keysight M8194A provides up to 4 independent RF channels and has a cost of approximately 300,000–400,000 \$, it can be easily understood that such a setup for generating the 8 different electrical channels falls well beyond what should be expected by a university research lab. The cost and complexity obviously increase even more dramatically for a 16×16 configuration, as it would require double the number of channels and significantly more advanced testing infrastructure. Beyond AWG requirements, additional high-bandwidth optical and electrical components such as clock generators for synchronization, high-bandwidth driver amplifiers for electrical amplification and custom RF probes for high-frequency signal delivery further contribute to the cost and of course the complexity of scaling up the experiment. This has been the main reason that we have “limited” our experimental demonstration to a maximum Xbar circuit dimension of 4×4 . However, even this circuit dimension provides meaningful insights into the scalability of our theoretical models taking into account that:

- the Xbar insertion loss and fidelity performance theoretical models have been validated using up to 4×4 Xbar prototypes, as reported in our previous Nature Communications article (M. Moralis-Pegios et al, “Perfect linear optics using silicon photonics”, *Nat Commun* 15, 5468 (2024). <https://doi.org/10.1038/s41467-024-49768-y>, reference [4] in our current manuscript),
- the Xbar noise and energy performance theoretical models were validated in a number of publications where 2×2 Xbar prototypes were utilized for analog matrix-vector multiplication tasks (see reference [13] in our original manuscript G. Giamougiannis et al., “Analog nanophotonic computing going practical: Silicon photonic deep learning engines for tiled optical matrix multiplication with dynamic precision,” *Nanophotonics* 12(5), 963 (2023), see also S. Kovaivos et al, “Scaling photonic neural networks: A silicon photonic GeMM leveraging a Time-Space multiplexed Xbar”, *IEEE/OSA J. on Lightwave Technol.*, vol. 42, no. 22, pp. 7825-7833, Nov. 2024)

We believe that all this previously published work confirms the consensus of the scientific community that Xbar circuit dimensions of up to 4x4 can reliably validate the respective theoretical models for higher Xbar scales, without requiring 8x8 or even 16x16 experimental prototypes.

Regarding the manufacturing challenges (crosstalk, cumulative losses) for large-scale integration please the reviewer can refer to our response on *Comments #2* and *#6* of the *Reviewer #1*.

Comment #2:

Enhance error rate analysis with noise source quantification and energy efficiency measurements. To quantify or evaluate the contributions of thermal noise, crosstalk, and nonlinear effects to SER/MER. To provide measured energy efficiency data (including laser and driver circuitry) for WDM vs. single- λ designs. It seems that the energy efficiency has not consider the optical-electrical conversion power.

Reply:

Regarding noise source quantification please refer to the response of Comment #4 of the 1st reviewer. On the other hand, non-linear effects in photonic systems mainly arise due to high optical power levels which in our system may lead to two photon absorption (TPA) in silicon waveguides [47] and/or self-heating effects in electro-absorption modulators (EAMs) [48]. TPA typically becomes significant at power levels above 10 mW [47], while self-heating in EAMs can occur when the incident optical power exceeds 6 mW [48]. Accordingly, we carefully monitor and control the optical power to ensure operation within safe and stable limits.

The calculation of energy efficiency for both single and WDM layouts includes lasers power, accounted for through wall-plug efficiency, as well as optical-to-electrical conversion via transimpedance amplifiers, as thoroughly analysed in Section 4 of the supplementary material. It is worth noting that the number of TIAs remains unchanged between the single and WDM layouts, as the 4λ -WDM design effectively introduces a capacity enhancement factor of $\lambda = 4$, as well as minimized losses and enhanced energy efficiency. As a result, a 128×32 layout using 4λ -WDM achieves the same word capacity as a single- λ 128×128 layout. The same principle is also valid for the driver circuitry (DACs) of the static EAMs employed for encoding the target vectors.

On the other hand, the driver circuitry of the EAMs used to encode the search vectors has indeed not been considered in our simulations, as the generation of the search vector is not part of the CAM table, but is instead employed in order to facilitate the proof-of-concept demonstration. This is fully consistent with the approach taken in the electronic CAMs [35], [36], [41], [42], where the energy efficiency is calculated based solely on the implementation of the CAM table.

Changes in the text: To address the reviewer's concern and provide clearer clarification on the energy efficiency comparison between the single- λ and WDM layouts, we have provided the following:

In supplementary material, section 4, line 268:

“This approach allows for minimization in the overall scale of the layout, allowing for a reduced number of output columns, thus reducing the splitting and combination required components achieving the same number of comparison operations. Based on that, due to the reduced splitting stages the required laser power launched into the crossbar array will be minimized, since less power will be needed in order to propagate the same power value at every column of the array. On top of that the component losses of the splitter/combiner stages will be also reduced, with the only additional loss parameter being the Mux and Demux stages that the different wavelengths will propagate through (3 stages for every column). Considering state-of-the-art low-loss multiplexer and demultiplexer components that can be utilized in the

WDM-architectures, a minimal losses of 0.2 dB per Mux/De-Mux module **Error! Reference source not found.**, can be estimated.

This leads to significantly reduced overall insertion losses for the WDM implemented schemes (IL_{WDM}) compared to the total losses of the conventional Xbar layouts ($IL_{single-wavelength}$), with the loss reduction being correlated to the number of allocated wavelengths that are used for the parallel comparisons and the reduced columns of every layout. Such an approach significantly reduces the overall insertion losses of the architecture, especially when scaling to 32 and 64-bit long comparison schemes.

The method of calculating the total losses of the conventional architecture ($IL_{single-wavelength}$) are thoroughly explained in **Error! Reference source not found.**, while the loss reduction parameters of the WDM layouts are provided below for every respective scale case of [4-128].”

Comment #3:

Update technical comparisons with state-of-the-art electronic CAM/TCAM benchmarks. To compare against recent CMOS advancements (e.g., 5nm-node TCAMs) in speed, power, and area efficiency.

Reply:

Please refer to the response that we have provided at *Comment#5* of the *1st Reviewer*.

Comment #4:

Clarify WDM engineering challenges and mitigation strategies. To detail wavelength stabilization methods and difficulty (e.g., thermal feedback loops) to counter temperature drift. What is the inter-channel crosstalk suppression (e.g., optimized filter bandwidths, guard bands)?

Reply:

The reviewer has made a quite accurate statement regarding the possible problems that can occur in our WDM implementation, regarding the utilised wavelength’s stabilization of the layout. The reviewer can see our respective reply, detailed in *Comment #7* of *Reviewer #1*, where we propose possible methods that can be used to stabilize and compensate the thermal drifting of ring resonator effects, as well as calibrate the multiplexer/demultiplexer elements, ensuring proper operation of the WDM scheme.

The reviewer can also our reply of *Comment #6* and *#7* of *Reviewer #1*, where we analyze the crosstalk response in our WDM layout.

Additional changes:

- The Data Availability section has been modified accordingly:

“The data cases that support the experimental results and findings of the study, as well as all the validation methods and procedures for the additional analysis performed have been provided with this paper during the submission process, with supplementary information files. In case of additional information and data of the study, complementary material can be provided upon request.”

- An additional Code Availability section has been added to the main manuscript:

“All software and source code files, as well as documentation and instructions for the analysis and validation of the experimental and theoretical findings of the study, have been provided with this paper, as

supplementary material and documents. Any additional specifications regarding the methods that have been followed can be provided upon request.”

- The References section has been updated with the following manuscripts:
 - “- *K. Huffman, W. C. & Pless, V. Fundamentals of Error-Correcting Codes, Chs. 1, 3 (Cambridge Univ. Press, 2003).*
 - *Hamming, R. W. Coding and Information Theory, Chs. 1–2 (Prentice-Hall, 1980).*
 - *Argiento, H. Filippi-Mazzola, E. Paci, L., Model-based clustering of categorical data based on the Hamming distance, arXiv:2212.04746.*
 - *Pinheiro A., Pinheiro Prisco H., et. al., The use of Hamming distance in bioinformatics. In Handbook of Statistics 28, 129–162 (Elsevier, 2012).*”
- Respective changes in the Supplementary material have been performed, according to the comments.

References:

- [1] E. Garzón, et al., "A Low-Energy DMTJ-Based Ternary Content- Addressable Memory With Reliable Sub-Nanosecond Search Operation," in IEEE Access, vol. 11, pp. 16812-16819, 2023, doi: 10.1109/ACCESS.2023.3245981
- [2] C. Wang, et al., "A Novel MTJ-Based Non-Volatile Ternary Content-Addressable Memory for High-Speed, Low-Power, and High-Reliable Search Operation," in IEEE Transactions on Circuits and Systems I: Regular Papers, vol. 66, no. 4, pp. 1454-1464, April 2019, doi: 10.1109/TCSI.2018.2885343.
- [3] X. Wang, et al., "A Novel Multi-Context Non-Volatile Content-Addressable Memory Cell and Multi-Level Architecture for High Reliability and Density," 2021 IEEE 10th Non-Volatile Memory Systems and Applications Symposium (NVMSA), Beijing, China, 2021, pp. 1-6, doi: 10.1109/NVMSA53655.2021.9628720.
- [4] G. Giamougiannis et. al., "A Coherent Photonic Crossbar for Scalable Universal Linear Optics," in JLT, vol. 41, no. 13 8, pp. 2425-2442, 15 2023.
- [5] Tae Joon Seok, Niels Quack, Sangyoon Han, Richard S. Muller, and Ming C. Wu, "Large-scale broadband digital silicon photonic switches with vertical adiabatic couplers," Optica 3, 64-70 (2016).
- [6] M. Pantouvaki et al., "Active Components for 50 Gb/s NRZ-OOK Optical Interconnects in a Silicon Photonics Platform," JLT, 35, 4, 2017.
- [7] C. Li et al., "A 3D-Integrated 56 Gb/s NRZ/PAM4 Reconfigurable Segmented Mach-Zehnder Modulator-Based Si-Photonics Transmitter," 2018 IEEE BiCMOS and Compound Semiconductor Integrated Circuits and Technology Symposium (BCICTS), San Diego, CA, USA, 2018
- [8] Agrawal, G. P. Fiber-Optic Communication Systems, 4th edn. 261 (John Wiley & Sons, 2012).
- [9] J.T. Koo, "Integrated-circuit content-addressable memories," IEEE Journal of Solid-State Circuits, volume: 5, issue: 5, pp.208-218, October 1970
- [10] BD.Yang,L.S.Kim,"A Low-Power CAM Using Pulsed NAND–NOR Match-Line and Charge-Recycling Search-Line Driver," IEEE J. Of Solid-State Circuits, Vol. 40, is. 8, pp.1736-1744, August 2005
- [11] B.-D. Yang, et. Al., "A Low Power Content Addressable Memory Using Low Swing Search Lines," IEEE Trans. on Circuits and Systems I: Regular Papers , vol. 58, no. 12, pp. 2849-2858, Dec. 2011

- [12] G. Kasai et al., "200 MHz/200 MSPS 3.2 W at 1.5 V V_{dd}, 9.4 Mbits ternary CAM with new charge injection match detect circuits and bank selection scheme" IEEE C. Int. Circuits Conf., 2003, pp. 387–390
- [13] T. Nagakartnik and J.R. Choi, "500-MHz high-speed, low-power ternary CAM design using selective match line sense amplifier in 65nm CMOS", 6th Inform. and Commun. Syst., pp. 60-63, Amman, 2015
- [14] E. Garzon, et. al. "A 128-kbit Approximate Search-Capable Content-Addressable Memory (CAM) With Tunable Hamming Distance," IEEE J. of Solid-State Circuits, article in press
- [15] K. Lee, et. al. "A 65-nm 0.6-fJ/Bit/Search Ternary Content Addressable Memory Using an Adaptive Match-Line Discharge," IEEE J. Solid-State Circuits, vol. 56, no. 8, pp.2574-2584, Aug. 2021
- [16] I. Arsovski, T. Hebig, D. Dobson, R. Wistort, "A 32 nm 0.58-fJ/Bit/Search 1-GHz Ternary Content Addressable Memory Compiler Using Silicon-Aware Early-Predict Late-Correct Sensing With Embedded Deep-Trench Capacitor Noise Mitigation," IEEE J. Solid-State Circ., vol.48 (4), 932-939, 2013
- [17] K. N ii, et. al. "A 28nm 400MHz 4-parallel 1.6Gsearch/s 80Mb ternary CAM," IEEE International Solid-State Circuits Conference Digest of Technical Papers, San Francisco, CA, USA, Feb. 2014
- [18] S. Jeloka, et. al. "A 28 nm Configurable Memory (TCAM/BCAM/SRAM) Using Push-Rule 6T Bit Cell Enabling Logic-in-Memory," IEEE Journal of Solid-State Circuits (Volume: 51, Issue: 4, pp. 1009-1021, April 2016,
- [19] Y. Tsukamoto, "1.8 Mbit/mm² Ternary-CAM macro with 484 ps Search Access Time in 16 nm Fin-FET Bulk CMOS Technology," IEEE Symposium on VLSI Circuits, Kyoto, Japan, June 2015
- [20] I. Avrovski, et. al. "1.4Gsearch/s 2Mb/mm² TCAM Using Two-Phase-Precharge ML Sensing and Power-Grid Pre-Conditioning to Reduce Ldi/dt Power-Supply Noise by 50%," IEEE International Solid-State Circuits Conference (ISSCC), San Fransisco, CA, USA, Feb. 2017
- [21] M. Yabuuchi, et. al. "12-nm Fin-FET 3.0G-search/s 80-bit × 128-entry Dual-port Ternary CAM," IEEE Symposium on VLSI Circuits, Honolulu, HI, USA, June 2018
- [22] M. Yabuuchi, et. al. "A 7nm Fin-FET 4.04-Mb/mm² TCAM with Improved Electromigration Reliability Using Far-Side Driving Scheme and Self-Adjust Reference Match-Line Amplifier," IEEE Sympos. VLSI Circuits, Honolulu, HI, USA, June 2020
- [23] C. Deshpande, et. al. "A 5nm Fin-FET 2G-search/s 512-entry x 220-bit TCAM with Single Cycle Entry Update Capability for Data Center ASICs," IEEE VLSI Symposium, Kyoto, Japan, 13-19 June 2021
- [24] S. Kumar, et. al., "A 3nm FinFET 2.2Gsearch/s 0.305fJ/b TCAM with Dynamically Gated Search Lines for Data-Center ASICs," IEEE Intern. Solid-State Circuits Conf, San Francisco, CA, USA, Feb. 2025
- [25] L.-A. Yu, et. al. "Design of the 2-nm Nanosheet NAND-type TCAM with High Speed and Compat Cell-size: 45% Layout-reduction of 3-nm TCAM," IEEE Silicon Nanoelectronics Workshop, Honolulu, HI, USA, June 2024
- [26] S. Pathak, P. Dumon, D. Van Thourhout and W. Bogaerts, "Comparison of AWGs and Echelle Gratings for Wavelength Division Multiplexing on Silicon-on-Insulator," in *IEEE Photonics Journal*, vol. 6, no. 5, pp. 1-9, Oct. 2014.
- [27] E. Timurdogan, et. al., "Automated wavelength recovery for microring resonators," in Proc. Conf. Lasers Electro-Opt., 2012, pp. 1–2.
- [28] K. Padmaraju, J. Chan, L. Chen, M. Lipson, and K. Bergman, "Thermal stabilization of a microring modulator," *Opt. Exp.*, vol. 20, no. 27, pp. 27999–28008, 2012.
- [29] K. Padmaraju, et. al., "Wavelength locking and thermally stabilizing microring resonators using dithering signals," *JLT*, vol. 32, no. 3, pp. 505–512, Feb. 2014.

- [30] F. Zanetto et al., "WDM-Based Silicon Photonic Multi-Socket Interconnect Architecture With Automated Wavelength and Thermal Drift Compensation," in *Journal of Lightwave Technology*, vol. 38, no. 21, pp. 6000-6006, 1 Nov.1, 2020
- [31] S.M. Lundberg, *et al.* Explainable machine-learning predictions for the prevention of hypoxaemia during surgery. *Nat Biomed Eng* 2, 749–760 (2018).
- [32] Y.C. Su, et.al., Visual Vocabulary Processor Based on Binary Tree Architecture for Real-Time Object Recognition in Full-HD Resolution, IEEE VLSI, 2000.
- [33] T. M. Chan, et. al., Approximating text-to-pattern Hamming distances, ACM SIGACT Symposium on Theory of Computing, STOC 2020, New York, NY, USA, 643–656.
- [34] C. Wang, et. al., Using Hamming Distance as Information for SNP-Sets Clustering and Testing in Disease Association Studies, 2015.
- [35] R. Mao, et al., Experimentally validated memristive memory augmented neural network with efficient hashing and similarity search. *Nat Commun* 13, 6284 (2022).
- [36] G. Karunaratne, et al., Robust high-dimensional memory-augmented neural networks. *Nat Commun* 12, 2468 (2021).
- [37] G. Pedretti, et al. Tree-based machine learning performed in-memory with memristive analog CAM, *Nat Comm.* 12, 5806 (2021).
- [38] K. Pagiamtzis and A. Sheikholeslami, "Content-Addressable Memory (CAM) Circuits and Architectures: A tutorial and survey," *IEEE Journal of Solid-state Circuits*, vol. 41, no. 3, pp. 712–727, Mar. 2006.
- [39] C. Minkenberg, *et al.*, "Co-packaged datacenter optics: Opportunities and challenges," *IET Optoelectronics*, vol. 15, no. 2, pp. 77–91, Mar. 2021.
- [40] R. S. Williams, "What's Next? [The end of Moore's law]," in *Computing in Science & Engineering*, vol. 19, no. 2, pp. 7-13, Mar.-Apr. 2017.
- [41] C. E. Graves, *et al.*, "Memristor TCAMs Accelerate Regular Expression Matching for Network Intrusion Detection," in *IEEE Transactions on Nanotechnology*, vol. 18, pp. 963-970, 2019.
- [42] Y. Goh, *et al.*, "High performance and self-rectifying Hafnia-based ferroelectric tunnel junction for neuromorphic computing and TCAM applications," 2021 IEEE International Electron Devices Meeting (IEDM), Dec. 2021
- [43] Huffman, W. C. & Pless, V. *Fundamentals of Error-Correcting Codes*, Chs. 1, 3 (Cambridge Univ. Press, 2003).
- [44] Hamming, R. W. *Coding and Information Theory*, Chs. 1–2 (Prentice-Hall, 1980).
- [45] Argiento, H. Filippi-Mazzola, E. Paci, L., Model-based clustering of categorical data based on the Hamming distance, [arXiv:2212.04746](https://arxiv.org/abs/2212.04746).
- [46] Pinheiro A., Pinheiro Prisco H., et. al., The use of Hamming distance in bioinformatics. In *Handbook of Statistics* 28, 129–162 (Elsevier, 2012).
- [47] Q. Lin, Oskar J. Painter, and Govind P. Agrawal, "Nonlinear optical phenomena in silicon waveguides: Modeling and applications," *Opt. Express* 15, 16604-16644 (2007).
- [48] D. Coenen et al., "Electro-Absorption Modulator Thermo-Optical Self-Heating Analysis," in *Journal of Lightwave Technology*, vol. 41, no. 18, pp. 6000-6006, 15 Sept.15, 2023.

Aristotle University of Thessaloniki

Department of Informatics
54124 Thessaloniki, Greece

Tel: +30 2310 990588

E-mail moschost@csd.auth.gr

3rd October 2025

Dear Reviewers,

We would like to thank the reviewers for their constructive feedback and thorough comments on our submitted manuscript in Nature Communications, entitled “**Nonlinear Optical Vector Processing using Linear Silicon Photonic Circuits for 50 Gb/s Memory and String Similarity functions**”, by T. Moschos, C. Pappas, S. Kovaivos, I. Roumpos, A. Prapas, A. Tsakyridis, M. Moralis-Pegios, C. Vagionas, Y. London, B. Tossoun, T. Van Vaerenbergh and N. Pleros, rendering it suitable for publication.

With this reply letter you can find attached our final changes and replies to the provided comments, along with the respective changes in the overall manuscript material.

Best regards,
Theodoros Moschos
Department of Informatics
Aristotle University of Thessaloniki

Revisions on the scientific journal of Nature Communications for the submitted manuscript of
“Nonlinear Optical Vector Processing at 50Gb/s using Linear Silicon Photonic Circuits”-
NCOMMS-25-11055A

Reviewer #1

From an academic and research point of view, for conceptual proof, small-scale architecture is acceptable. The authors have answered all the questions from the three reviewers, and I believe the answers are accurate. There are no further comments.

Reply:

We would like to sincerely thank the reviewer for the positive evaluation on our proposed architectures. We are glad that our responses have addressed all comments and questions. The constructive feedback that has been provided through the review process has helped us improve and strengthen the clarity and overall quality of the current manuscript.

Reviewer #2

Comment:

I thank the authors for addressing some of my concerns. In response to my comment #2, the authors provided “Table 2: State-of-the-art electronic TCAMs” in the revised supplementary document. While it is true that the chip in this work operates at a significantly higher clock frequency (50 GHz) than electronic TCAMs, its energy efficiency (energy/search/bit), array size, and area per cell are significantly worse than those of its electronic counterparts. For example, compared with Ref. 26, which uses an outdated CMOS technology node of 32 nm, the area/cell in this work is 113,095 times larger and the energy/search/bit is 55 times higher; compared with Ref. 34, which uses an advanced CMOS node of 3 nm, the area/cell in this work is 472,637 times larger and the energy/search/bit is 105 times higher.

More importantly, it is nearly impossible that the array size of a photonic circuit can reach the same scale as that of electronic TCAMs, which can easily achieve 512×220 . The array size is an important parameter that cannot be ignored in TCAM applications. Lightmatter and Lightelligence have indeed presented large-scale photonic chips for matrix multiplications (128×128 and 64×64 , respectively). However, the operating clocks of their photonic chips are 2 GHz and 1GHz, respectively. I am not convinced that each modulator can still be operated at 40 GHz in a practical scenario when the photonic circuit demonstrated in this work is scaled up (for example, to 128×128).

Therefore, while I think this work provides a novel approach, I do not think that the results presented in this work represent a significant advancement compared to electronic TCAMs. Thus, I cannot recommend its publication in Nature Communications.

Reply:

We would like to thank the reviewer for the effective criticism and comments on our presented work, and also for recognizing the novelty of our approach. At first, we are delighted that we managed to address a part of the reviewers’ comments. We completely understand the reviewer’s concern, mentioning a detailed comparison and highlighting the maturity of the electronic TCAM technologies compared to our optical one.

We fully agree that in terms of energy/search/bit as well as area/cell efficiency the current electronic CMOS TCAMs, after decades of maturity and optimization across their nodes, remain superior compared to the presented photonic layouts. However, our current contributed work does not lie in matching CMOS at its maturity level, but in demonstrating a fundamentally new direction where photonics offer some key advantages:

- **Bandwidth & Parallelism:** Our prototypes operate at clock rates of 50 GHz, i.e. 25x higher than current electronic TCAMs, offering also parallel operations via wavelength multiplexing. This parallelism is not available to electronic implementations, and it becomes increasingly critical as system I/O line rates move toward multi-Tb/s.
- **Interconnect Bottleneck Relief:** In modern systems, it is revealed that the dominant energy cost is actually data movement and not storage of the information itself. Photonic circuits naturally provide low-latency and low-energy interconnects, which scale favourably in energy-delay product compared to electronics at multi-Gb/s per lane operations.
- **Passive Computation:** Unlike CMOS TCAMs, which rely on sequential transistor switching and complex interconnects, our schemes perform all the respective computations via passive interference, with delays mostly determined by the optical path length. This results in intrinsic ps-scale latency that is independent of logic depth.

By comprehending all the above characteristics, our presented coherent photonic prototypes showcase promising energy requirements as well as latency-free interference processes. Moreover, apart from the high clock rate of our implementations our proposed WDM layout enables a parallel computational approach, offering multi-dimensional operations without necessarily requiring high-scale photonic arrays, similar to the respective electronics. This allows for a reasonable node array, allowing for realistic photonic structures that can efficiently be controlled retaining their functionality across their scale and also remain into a reasonable energy cost in the cases of required scalability, even at high data rates, as highlighted by our additional analysis.

Finally, we would like to emphasize that silicon photonics, while still being less mature than CMOS, undergo very rapid technological progress [1]. Over the past decade, we have witnessed dramatic improvements in footprint reduction, wafer-scale integration, heterogeneous integration with III-V materials and advanced packaging. These trends are very similar to the evolutionary trajectory that electronic TCAMs themselves experienced over several decades. Given this continuous progress, we are confident that the density and energy efficiency gap with CMOS will gradually shrink, while maintaining the unique advantages of photonics in terms of bandwidth scaling, interconnect efficiency, and parallelism, rendering also our proposed prototypes efficient in larger-scale implementations.

Additional changes:

Based on the editorial requests respective text formatting across all the main manuscript and supplementary material of the presented work have been modified. Additionally, the respective manuscript figures have been modified and changed in order to comply with the respective requests.

References:

- [1] N. Margalit et al., “Perspective on the future of silicon photonics and electronics,” *Appl. Phys. Lett.* 118, 220501 (2021).